# Comparison of two automated aerosol typing methods and their application on an EARLINET station.

Kalliopi Artemis Voudouri[1], Nikolaos Siomos[1], Konstantinos Michailidis[1],
Nikolaos Papagiannopoulos[2,3], Lucia Mona[2], Carmela Cornacchia[2], Doina Nicolae[4], and Dimitris Balis[1]

[1]Laboratory of atmospheric physics, Physics Department, Aristotle University of Thessaloniki, Greece
[2]Consiglio Nazionale delle Ricerche, Istituto di Metodologie per l'Analisi Ambientale (CNR-IMAA), Tito Scalo (PZ), Italy
[3]CommSensLab, Dept. of Signal Theory and Communications, Universitat Politècnica de Catalunya, Barcelona, Spain
[4]National Institute of R&D for Optoelectronics (INOE2000), Magurele, Romania

**Correspondence:** Kalliopi Artemis Voudouri (kavoudou@auth.gr)

**Abstract.** In this study we apply and compare two algorithms for the automated aerosol type characterization of the aerosol layers derived from Raman lidar measurements over the EARLINET station of Thessaloniki, Greece. Both automated aerosol type characterization methods base their typing on lidar derived aerosol intensive properties. The methodologies are briefly described and their application on three distinct cases is demonstrated and evaluated. Then the two classification schemes were applied in the automatic mode on a more extensive dataset. The dataset analyzed corresponds to ACTRIS/EARLINET (European Aerosol Research Lidar NETwork) Thessaloniki data acquired during the period 2012-2015. Seventy one layers out of 110 (percentage of 65%) were typed by both techniques and 56 of these 71 layers (percentage of 79%) were attributed to the same aerosol type. However as shown the identification rate of both typing algorithms can be changed regarding the selection of appropriate threshold criteria. Four major types of aerosols are considered in this study: Dust, Maritime, PollutedSmoke and CleanContinental. The analysis showed that the two algorithms, when applied to real atmospheric conditions, provide typing results that are in a good agreement regarding the automatic characterization of PollutedSmoke, while there are some difference between the two methods regarding the characterization of Dust and CleanContinental. These disagreements are mainly attributed to differences in the definitions of the aerosol types between the two methods, regarding the intensive properties used and their range.

## 1 Introduction

Aerosol classification is a key parameter in understanding the impact of different aerosol sources on climate, weather systems, and air quality. Given that aerosols are emitted from various sources, a significant number of aerosol types with different optical and microphysical properties coexist in the atmosphere. As a consequence, considerable uncertainties have been raised in the quantification of aerosols in the radiative transfer calculations and their interactions with clouds, leading the interest of wide communities to improving common understanding related to aerosol types and sources.

Nowadays, aerosol typing is an interesting topic recognized at international level, as inferring the aerosol source is an important tool for identification of events, alerting of dangerous situations, definition of mitigation strategies. The International

Satellite Aerosol Science Network (AEROSAT), for example, is an initiative established from different research groups around the world focusing on satellite aerosol retrieval. AEROSAT identifies the need of a common background for aerosol typing and typing procedures comparison efforts as one of the main important task to be tackled into the aerosol community (Mona et al., 2015).

Thus, a number of typing schemes have been applied to classify aerosols, based on the synergistic information of the backscattering and extinction coefficients and the polarization state of the received light at different wavelengths (e.g, Ansmann and Müller, 2005, Groß 2013). For example, Hamill et al. (2016), developed an aerosol classification scheme using the optical properties derived from the AERONET (Aerosol Robotic Network) ground-based network of sun photometers. Five types of aerosols were defined based on reference clusters, characteristic of a particular type of aerosol (i.e., Maritime, Urban Industrial,

Dust, Mixed and Biomass Burning). Schmeisser et al. (2017) also, used aerosol optical properties (i.e., scattering Ångström exponent, absorption Ångström exponent and single scattering albedo) derived from in situ measurements in order to classify aerosols. Their typing scheme was based on three different aerosol classification methods.

Active remote sensing instruments such as ground or space-based lidars, are a key technique for characterizing aerosols as they can provide vertically-resolved information of extensive (e.g., aerosol backscatter coefficient, aerosol extinction coefficient

and volume depolarization ratio) and intensive properties (e.g., Ångström exponent, lidar ratio and particle depolarization ratio) of different aerosol types. The extensive properties depend on the aerosol concentration, whilst intensive ones are type sensitive and provide separate classification for each detected layer (e.g., Müller et al., 2007; Ansmann et al., 2011; Tesche et al., 2011; Burton et al., 2012; Pappalardo et al., 2013; Groß et al., 2013; Amiridis et al., 2015; Giannakaki et al., 2015; and Baars et al., 2016). Nevertheless, even intensive properties might not be sufficient to guarantee accurate typing, as some aerosol types

(e.g., biomass burning and industrial pollution) have very similar intensive properties but are attributed to different sources and generating mechanisms. Indeed, volcanic and desert dust particles are both characterized by the asphericity of the particles, observable and measured through the linear depolarization ratio (while typically the other aerosol types have small values of this parameter) and also, their intensive optical properties could be similar in ranges e.g., Ångström exponents and lidar ratio (see for example Fig. 5 in Nicolae et al., 2018; Fig. 7 in Papagiannopoulos et al., 2018 and the variety of volcanic values in

Mona et al., 2012).

Aerosol typing schemes have been developed for high-resolution lidar measurements of space-borne lidars (e.g., CALIPSO, Cloud-Aerosol Lidar and Infrared Pathfinder Satellite Observation, Omar et al. 2009; EarthCARE, Illingworth et al. 2015; CATS, Yorks et al., 2016), airborne - High Spectral Resolution Lidar -HSRL- measurements (e.g., Burton et al., 2012; Groß et al., 2013) and multiwavelength Raman measurements (e.g., Nicolae et al., 2018; Papagiannopoulos et al., 2018). The CALIPSO

mission uses a decision-tree based on lidar profiles and external data (Omar et al., 2009) in order to classify the aerosol load in ten aerosol subtypes, i.e., marine, dust, polluted continental / smoke, clean continental, polluted dust, elevated smoke, dusty marine, PSC aerosol, volcanic ash, sulfate/other. The upcoming mission of EarthCARE will depend on the high-spectral-resolution lidar products for aerosol classification. The aerosol intensive properties at 355nm combined with observations of the 355nm aerosol optical depth will constitute the basic input for the aerosol-type determination (Wandinger et al., 2016). An

aerosol classification scheme from HSRL aerosols measurements has been introduced by Burton et al. (2012). Their technique

applies an objective multivariate analysis using lidar intensive properties (i.e., the particle linear depolarization ratio at 532 nm, the particle lidar ratio at 532 nm, the backscatter-related 532 -to- 1064nm color ratio, and the ratio of particle linear depolarization ratios at 1064 and 532 nm) and is able to discriminate 8 aerosol types: smoke, fresh smoke, urban, polluted maritime, maritime, dusty mix, pure dust and ice.

Several other approaches have been developed based on the information provided by multispectral data of ground based measurements. EARLINET, for example, is a well structured network of advanced laser remote sensing stations with the main purpose of understanding the horizontal and vertical distribution of aerosols on the European scale. The EARLINET data have been extensively used not only for climatological studies (e.g., Matthias et al., 2004; Mattis et al., 2004; Amiridis et al., 2005), but also for studies on Saharan dust outbreaks (e.g., Ansmann et al., 2003; Balis et al., 2004; Papayannis et al., 2008; Tesche
et al., 2009b; Mona et al., 2014; Binietoglou et al., 2015), volcanic eruptions (e.g., Pappalardo et al., 2004; Sawamura et al., 2012; Pappalardo et al., 2013) and biomass burning events (e.g., Balis et al., 2003; Tesche et al., 2011; Nicolae et al., 2013).

Efforts for detailed knowledge of the aerosol sources have been conducted in the framework of EARLINET, focusing on the aerosol characterization and typing. For example, Müller et al. (2007) presented for the first time a statistical analysis of lidar ratios for almost all climatically relevant aerosol types based solely on Raman lidar measurements. The analysis covered the
most important aerosol types such as maritime particles, desert dust particles and aged biomass-burning smoke. Lidar depolarization measurements have also been used to differentiate between different types of aerosols, since they consitute an indicator of the sphericity of particles. Tesche et al. (2009) separate the optical properties of desert dust and biomass burning particles using multiwavelength Raman and depolarization measurements. Methods for fine and coarse mode separation for dust outbreak cases, using the 532 nm particle depolarization ratio have also been developed (e.g., Mamouri and Ansmann, 2014).
Additionally, combined measurements of the lidar ratio from ground-based Raman lidars along with aerosol depolarization values and the size-sensitive Ångström exponent, were proved to be a useful tool for the separation of aerosol types as shown by Groß et al. (2011). Furthermore, the combined use of lidar observations and transport model simulations permit the discrimination of desert dust and volcanic ash particles that typically, have the same optical characteristics. Simultaneous observations of desert dust and ash particles were made during the Eyjafjallajökull volcanic eruption in 2010 and the methodology for the
type discrimination was presented by Papayannis et al. (2012); Mona et al. (2012); and Pappalardo et al. (2013).

Automated aerosol typing schemes have been discussed to be implemented in the next version of the EARLINET Single Calculus Chain tool (SCC; D'Amico et al., 2015). SCC was developed for the analysis of the data of different lidar systems in an automated, unsupervised way and currently SCCv5.0.11 delivers profiles of optical aerosol properties. Future work is dedicated to the implementation of new features like profiles of intensive optical properties, detection of aerosol layer geometrical
properties and calculation of the intensive optical properties per detected layer, which will further allow the classification of the observed layers into aerosol types. Therefore, the need for defining a common background on aerosol types (even nomenclature is not harmonized nowadays) and for comparing aerosol types (which is not a numerical value) derived by different methods is on the top of interest and the following questions have been raised:

- Is there a common understanding for the aerosol types between different typing schemes?

- Can automated methods provide operationally accurate typing in automatic mode when applied in data stored in a climatological database?

- How can we compare different typing methodologies? Which parameters affect the agreement or disagreement between them and the ability to provide typing results.

Given the aforementioned status, a first attempt of comparing and evaluating two classification tools developed within EARLINET, that provide near-real-time aerosol typing information for the lidar profiles of Thessaloniki, is presented. Our aim is: (i) to check the performance of both supervised learning techniques in their low resolution mode, which is the case for the majority of the available measurements within EARLINET, when applied to lidar data from a station where, typically, variable mixtures of aerosols are present and (ii) to investigate the reasons of typing agreement and disagreement with respect to the uncertainties and the threshold criteria applied. This paper can be considered as a step towards the general objective of finding translating rules and way for quantifying differences between typing procedures.

The article is structured as follows: the Thessaloniki EARLINET lidar station is presented in Sect. 2. The two automatic aerosol classification methods and the methodology used to characterize the layers in four basic aerosol types (i.e., Dust, PollutedSmoke, Maritime, CleanContinental) are presented in Sect. 3. In Sect. 4 the accuracy of the algorithms is tested by comparison with pre-classified case studies, the evaluation for the whole period under study is presented and the results of the study are discussed. Finally, Sect. 5 contains the summary and the conclusions of this article.

## 2   Lidar system and measurement site

The Thessaloniki lidar system (THELISYS) is operated and maintained by the Laboratory of Atmospheric Physics that is located in the Physics Department of the Aristotle University of Thessaloniki ($40.5^oN, 22.9^oE, 50m$). THELISYS is used for the detection of aerosol particles as a part of EARLINET (Bosenberg et al., 2003; Pappalardo et al., 2014) since 2000 and now is part of the Aerosols, Clouds, and Trace gases Research InfraStructure (ACTRIS; www.actris.eu/). Systematic measurements are performed, following EARLINET's schedule (i.e., every Monday morning, and every Monday and Thursday evening after the sunset). Additional measurements are performed during Saharan dust outbreaks, smoke advection from biomass burning, volcanoes eruption, and CALIPSO correlative measurements. The current setup of THELISYS includes three elastic backscatter channels at 355 nm, 532 nm and 1064 nm, two nitrogen Raman channels at 387 nm and 607 nm and two depolarization channels. These two channels have been added to measure the cross and parallel polarized signal at 532 nm, but due to technical issues the particle depolarization ratio is available after August 2018. A detailed description of THELISYS can be found in Siomos et al. (2018a) and Siomos (2018c). Data from THELISYS are regularly analysed and quality assured and are publically available at www.earlinet.org.

Thessaloniki is in a location where many different types of aerosols coexist (Amiridis et al., 2009; Giannakaki et al., 2010; Siomos et al., 2018a). Dust events are dominant during summer above 1.5 km and in autumn below 1.5 km as shown by Siomos et al. (2018a). Marinou et al. (2017), also, used CALIPSO data and confirmed the existence of dust plumes during advection

episodes over 2 km in summer. Similarly, the most intense biomass burning episodes tend to occur during summer in the free troposphere and are probably associated to wildfires rather than agricultural fires that tend to be predominant during spring and autumn (Siomos et al., 2018). Continental layers observed over Thessaloniki station are attributed to mixtures of anthropogenic pollution and particles from natural sources and even mixtures of maritime aerosol. Therefore, Thessaloniki is well suited for

aerosol typing studies and for the investigation of the performances of different aerosol typing algorithms.

## 3   Aerosol Typing Methods

The two automatic aerosol typing methods require only lidar data with $3\beta + 2\alpha$ configuration (3 backscatter and 2 extinction coefficient profiles) without any use of ancillary external information. Specifically, the typing methods make use of the available aerosol type-sensitive intensive properties. As multiwavelength Raman lidars have the ability to measure directly aerosol

extinction and aerosol backscatter coefficient profiles in several wavelengths, a number of intensive properties can be aquired with high accuracy. A number of the obtained quantities do not depend on the aerosol load but they can be linked to the size, the chemical composition, and/or the asphericity of the particles. The investigation of these quantities is important to infer the aerosol type as discussed in many papers (e.g. Burton et al., 2012; Groß et al., 2013; Wandinger et al, 2016). In general, the ability for a succesfull classification of the different aerosol types and mixtures depends on the measurments uncertainties.

As large measurement uncertainties prevent a correct aerosol type separation, high quality measurements are mandatory. The EARLINET lidar data follow the quality standards that have been established, in order to make the lidar products of the different systems comparable and to be able to provide quality-assured datasets of network products (Freudenthaler et al., 2018). So, the lidar dataset used in this study can be considered for typing characterization.

The intensive properties relevant to this study are: the extinction-related Ångström exponent (AE), the backscatter-related

Ångström exponent (BAE), the ratio of the backscatter coefficients profiles (color ratios - CR), the lidar ratio (LR), and the ratio of the lidar ratios (RLR). The respective formulas are provided in the following equations, where $\lambda$ is the wavelength, z is the height, a is the aerosol extinction coefficient, and b is the aerosol backscatter coefficient.

$$AE_{\lambda1/\lambda2}(z) = -\frac{ln(\frac{a(\lambda_1,z)}{a(\lambda_2,z)})}{ln(\frac{\lambda_1}{\lambda_2})} \tag{1}$$

$$BAE_{\lambda1/\lambda2}(z) = -\frac{ln(\frac{b(\lambda_1,z)}{b(\lambda_2,z)})}{ln(\frac{\lambda_1}{\lambda_2})} \tag{2}$$

$$CR_{\lambda1/\lambda2}(z) = \frac{b(\lambda_1,z)}{b(\lambda_2,z)} \tag{3}$$

$$LR(\lambda_1,z) = \frac{a(\lambda_1,z)}{b(\lambda_1,z)} \tag{4}$$

$$RLR_{\lambda2/\lambda1}(z) = \frac{LR(\lambda_2,z)}{LR(\lambda_1,z)} \tag{5}$$

The aerosol backscatter coefficients at two wavelengths ($\lambda_1 < \lambda_2$) are combined to give a backscatter-related aerosol Ångström exponent – BAE($\lambda_1/\lambda_2$)-. This quantity provides information about the aerosol size, and in contrast to the extinction coefficient, is associated with smaller uncertainty. This is attributed mainly to the derivative in the inversion algorithm (e.g., Ansmann et al., 2002; Freudenthaler et al., 2009), which produce larger uncertainties, while the backscatter coefficient calculated from the combination of Raman-elastic channels is less sensitive. The ratio of the aerosol extinction to backscatter coefficient is called lidar ratio – LR$\lambda_1$ – and changes largely for aerosols with different chemical and physical properties. This quantity has been valuable for aerosol characterization as demonstrated in previous studies (Müller et al., 2005; Mattis et al., 2004). The ratio of the lidar ratios – LR$\lambda_1$/LR$\lambda_2$ – can be used to assess the spectral dependency of the different aerosol types.

It is worthwhile to mention that the particle linear depolarization ratio is an intensive property that effectively discriminates spherical and non-spherical particles in the atmosphere. Nevertheless, the particle depolarization can be very sensitive to the calibration procedure, and the values of the calculated particle depolarization introduce also, a significant uncertainty. Given that, this quantity is not used in the aerosol typing presented here, this study assesses aerosol typing capabilities for $3\beta + 2\alpha$ lidars that do not make depolarization measurements.

### 3.1    Neural network Aerosol Typing Algorithm based on Lidar data - NATALI

The NATALI (Neural network Aerosol Typing Algorithm based on Lidar data) software relies on Artificial Neural Networks (Nicolae et al., 2018). The development of this tool started in the framework of EARLINET, with the main purpose of identifying the most probable aerosol type using a combination of mean-layer intensive optical parameters (i.e., lidar ratios, Ångström exponent, color ratios) from the provided aerosol backscatter and extinction coefficient profiles of lidar systems, without any additional information. The NATALI software consists of three independent, but inter-connected modules: the input, the typing and the output module. The input module requires optical properties profiles as those measured by EARLINET stations, namely the aerosol extinction coefficient and the aerosol backscatter coefficient profile. Optionally, the linear particle depolarization profile at 532 nm can be provided, so as to allow a better classification and to increase the number of classified aerosols types.

In a first step, the typing module identifies the geometrical boundaries of the layers by applying the gradient method on the 1064 nm backscatter coefficient profile (Belegante et al., 2014). For every detected layer from the input module, calculations of the mean layer values of the intensive optical parameters and the associated uncertainties are performed (Nicolae et al., 2016). The extinction-related Ångström Exponent at 355nm and 532nm ($AE_{355/532}$), the backscatter-related Ångström Exponent at 355nm and 532nm ($BAE_{355/532}$), the backscatter-related Ångström Exponent at 532nm and 1064nm ($BAE_{532/1064}$), the lidar ratio at 355 nm ($LR_{355}$), the lidar ratio at 532 ($LR_{532}$), and the ratio of the backscatter coefficients profiles (color ratios) $CR_{355/532}$ and $CR_{532/1064}$, are calculated. The acceptable ranges for the calculated intensive optical parameters should be between acceptable limits (i.e., extinction-related Ångström Exponent values, backscatter-related Ångström Exponent values and color ratio values between -2 and 6 and lidar ratio values between 5 and 200), so as the quality of the input data to be ensured.

In a second step three Artificial Neural Networks (ANN) are interrogated simultaneously regarding the aerosol type using the aforementioned optical properties calculated for each layer. The ANNs have been trained using synthetic data from a specially designed aerosol model. Specifically, the synthetic database was developed using the aerosol model builted for 350, 550, and 1000 nm sounding wavelengths, based on the 61 wavelengths of OPAC (Optical Properties of Aerosols and Clouds) software package (Hess et al., 1998), for which the microphysical characteristics of the aerosols are available from the Global Aerosol Data Set (GADS). These wavelengths are then re-scaled to the usual lidar wavelengths (i.e., 355, 532 and 1064 nm) using an average Ångström exponent equal to one. This was considered a valid assumption for all aerosol types, taking into account the small difference between the lidar and the model wavelengths. A comprehensive description of the developed aerosol model can be found in Nicolae et al., (2018).

The identification of the most probable aerosol type is then made through a voting procedure, using the results from the three ANNs interrogated. Over 50000 aerosol synthetic data have been used to train the ANN and identify the better ANNs to classify the aerosols type from multiwavelength lidar data. The capability of ANNs to resolve the overlapping clusters of the intensive optical parameters is used on NATALI algorithm. The answer is selected based on a statistical approach. The selected types of ANNs classify the aerosols based on the response with high: i) confidence (i.e. the probability of having one of the aerosol types) and ii) stability over the uncertainty range (i.e., the percentage of agreement for values between error limits). Therefore, answers with low confidence are filtered out and NATALI returns the 'Unknown' type. These threshold criteria are selected by the user and are expressed as percentages. Higher values of these criteria, enhance the confidence of the classification and affect the identification rate of the typed cases. In our study, we test their application and we define the confidence level for the output retrievals higher than 90% (minimum accepted confidence) and the minimum agreement threshold as default (i.e., 25%), in order to check the performance of NATALI in the highest confidence level.

Depending on the availability of the particle linear depolarization ratio and the quality of the provided lidar profiles, the derived typing can be either of high resolution (AH), or low resolution with depolarization (AL) or low resolution without depolarization (BL). Pure aerosols categories, and even mixtures of three aerosols types can be obtained from the NATALI algorithm. In the high resolution typing, 14 aerosol types can be distinguished (i.e., Continental, ContinentalPolluted, Dust, Maritime/CC, Smoke, Volcanic, Coastal, CoastalPolluted, ContinentalDust, ContinentalSmoke, DustPolluted, MaritimeMineral, MixedDust and MixedSmoke) when the quality of the provided optical products is high enough. In the low resolution typing (AL), 6 predominant aerosol types can be provided but with high uncertainty (i.e, Continental, ContinentalPolluted, Smoke, Dust, Maritime and Volcanic). The low resolution typing (BL) provides 5 predominant aerosol types (i.e., Dust, ContinentalPolluted, Smoke, Continental, Maritime) either pure or mixed, when the depolarization information is not provided. Finally, the output module provides the intensive optical parameters within each layer along with their mean value and the corresponding uncertainty. The complete and detailed typing procedure derived by NATALI can be found at Nicolae et al., (2016) and at Nicolae et al., (2018).

Application of NATALI on EARLINET data samples was also conducted. Observational data from the EARLINET-CALIPSO database and measurements derived by a Raman depolarization Lidar from the Romanian National Institute for Research and Development in Optoelectronics (Belegante et al., 2011) were used. The comparison between the NATALI aerosol typing mod-

ule and these observational data showed consistent results (Nicolae et al., 2018). This comparison on EARLINET data samples showed the capability of NATALI to retrieve the aerosol type from an extensive dataset, with variable quality and physical content.

## 3.2 EARLINET Mahalanobis distance-based typing algorithm - EMD

The EARLINET Mahalanobis distance-based typing algorithm is a method specifically developed for the use on the EAR-LINET database with a high level of flexibility in order to adapt to the different lidar set-ups and needs (Papagiannopoulos et al., 2018). The algorithm applies the Mahalanobis distance classifier (Mahalanobis, 1936) to classify observations into maximum 8 (Dust, Volcanic, MixedDust, PollutedDust, CleanContinental, MixedMarine, PollutedContinental, Smoke) and minimum 4 (Dust, Maritime, PollutedSmoke, CleanContinental) aerosol classes, considering the needs of each user and the

provided number of the intensive properties. The method demonstrated to be high performing on aerosol typing using optical properties measurements in previous studies (e.g., Burton et al., 2012; Russell et al., 2014; Hamill et al., 2016).

The classification of each aerosol layer is made by calculating the distance of an observation from the predefined reference classes and by attributing each observation to a specific class based on the minimum distance. Given that the overall predictive accuracy of a typing algorithm depends on the predefined reference classes, the choice of the appropriate reference dataset is

a crucial parameter. So, a well-characterized EARLINET dataset of observations from 2008 to 2010 was used to define the aerosol classes (Pappalardo et al. 2013; Papagiannopoulos et al. 2016a, and Schwarz 2016). As this dataset did not include all the typical aerosol components presented over Europe (i.e., Continental, Marine, Mineral dust and dust mixtures, Smoke, Volcanic ash), the observations were enhanced with additional ones (already published in literature), contributing to a total of 69 layers (Pappalardo et al., 2013; Papagiannopoulos et al., 2016a).

In a next step, a sensitivity analysis was performed in order to identify the classifying properties that provide the most adequate information to better predict of the correct aerosol class. The intensive properties that held the most weight among others in the classification were the backscatter-related Ångström Exponent at 355nm and 1064nm ($BAE_{355/1064}$), the lidar ratio at 532 nm ($LR_{532}$), and the ratio of the lidar ratios ($LR_{532}/LR_{355}$).

Consequently, the initial reference dataset was split into a training and a validation dataset, in order to evaluate the predictive

accuracy of the automatic method. Therefore, the training dataset was used to calculate the classification functions and then each observation was validated using the training dataset. Additionally, the effect of the depolarization information to the training phase was also investigated. To complement the reference dataset in this context, general literature values for particle linear depolarization ratio at 532 nm were used. Generally, the depolarization ratio is a really important parameter that can strengthen the typing procedure in the presence of aspherical layers, but Papagiannopoulos et al. (2018) showed that for the

5 (Dust, Maritime, PollutedContinental, Smoke, CleanContinental) and 4 (Dust, Maritime, PollutedSmoke, CleanContinental) classes, the particle linear depolarization ratio became less important (in this case the highest weight in the classification corresponds to the lidar ratio at 532 nm).

Finally, the assessment of the predictive performance of the algorithm was tested on a testing dataset. For this purpose, EARLINET data collected during the ACTRIS Summer 2012 intensive measurements (Sicard et al., 2015; Granados-Muñoz et

al., 2016b) were chosen to test the automatic typing algorithm. The testing dataset comprised of 47 layers, 21 of which yielded depolarization ratio values. The performance of the algorithm was checked for each of the grouping classes (i.e., 8,7,6,5,4) and the predictive accuracy of the algorithm increased up to 90% when the aerosol classes that tend to reflect the same optical properties values were combined into 4 (Dust, Maritime, PollutedContinental + Smoke, CleanContinental) without providing the information of the depolarization. The study concluded that the fewer aerosol classes (i.e., 4, 5, 6 classes) could provide a successful prediction accuracy, even without depolarization values, but, nonetheless, a coarser and less insightful classification. Dust classification showed a high success rate, whilst the aerosol types that performed worse were the smoke and polluted continental aerosol. However, when these two categories were combined into a single aerosol class, the correct prediction increased. More detailed description about the algorithm, the reference dataset and the set of the intensive parameters defined to separate different aerosol types can be found in Papagiannopoulos et al. (2018).

The output of the algorithm are: (i) the calculated Mahalanobis distance of the layers from each aerosol class and (ii) the calculated chi-square probability. For improving the reliability of the output, two screening criteria are applied to the calculated distances following the procedure of Burton et al. (2012). The first one has to do with the minimum accepted distance which depends on the number of degrees of freedom (i.e., the type dependent intensive properties) and the second one has to deal with the equal distances. In our study, we use the aforementioned 3 classifying parameters (i.e., the backscatter-related Ångström Exponent at 355nm and 1064nm ($BAE_{355/1064}$), the lidar ratio at 532 nm ($LR_{532}$), and the ratio of the lidar ratios ($LR_{532}/LR_{355}$)) and the minimum accepted distance for a measurement to be typed is 4. When the distance is higher than the defined threshold, which means that no similarity with the aerosol classes of the reference datasets is found, the observation is not typed. So, the first filter ensures that the calculated distance has 99% probability (cumulative probability) to belong to the class that was estimated. If this happens for more than one classes, then the second filter is applied. According to this one, the normalized probability (the probability based on the calculated distances) of the aerosol class needs to be higher than 50% (Papagiannopoulos et al., 2018). Overall, the application of the two screening criteria, enhances the confidence of the classification. However, the selection of a strict threshold (i.e., normalized probability greater than 60%) enhances the correct prediction, but reduces the identification rate of the algorithm.

### 3.3 Methodology

Fifty four (54) Raman lidar cases of aerosol measurements (backscatter coefficient profiles at 1064nm, 532nm and 355nm, as well as the extinction coefficient profiles at 532nm and 355nm) over Thessaloniki during the period 2012-2015 were used for this study. These input parameters were processed with NATALI algorithm for the identification of the layer boundaries, the calculation of their mean intensive optical parameters and their corresponding uncertainties. The NATALI typing was performed in the low resolution typing configuration (5 predominant aerosol types - Dust, Smoke, Continental Polluted, Continental and Maritime) since particle linear depolarization ratio measurements for Thessaloniki were not available for the study period.

In what follows, we merged the output types from NATALI that tend to reflect the same aerosol characteristics, and hence we evaluate the corresponding effects on the typing rate of the algorithms. Thus, the smoke and the polluted continental categories were grouped into the more generic type of small particles with high lidar ratio values (PollutedSmoke). The selection of four

main aerosol classes stems from the availability of intensive properties, the difficulty in deriving a confident classification without particle linear depolarization ratio and the difficulty in discriminating polluted continental and smoke particles that reveal the same type characteristics. Regardless, the aerosol classes describe the major aerosol components. The identified layer boundaries from NATALI are used as input in the EARLINET Mahalanobis distance-based typing algorithm. Considering the aforementioned typing merging, the EARLINET Mahalanobis distance-based typing algorithm was set to classify observations into 4 aerosol classes: CleanContinental, Dust, Maritime and PollutedSmoke.

Table 1 lists the aerosol types used in the aerosol classification and their correspondence with the available types provided by the two typing methods and Table 2 lists the mean aerosol optical properties of the reference aerosol types defined by the two algorithms. The idea here is to compromise: i) the resolution (low) of the automatic classification owing to the availability of the optical properties (i.e., 3+2 lidar configuration), and ii) the type definition, which does include the wide spectrum of the aerosol types provided by the two automated typing techniques. As it is evident from the values presented in Table 2, the lidar classification scheme consists of the main classes: (i) large particles with medium lidar ratios (i.e., dust-like particles), (ii) large particles with low lidar ratios (i.e., maritime particles), (iii) small particles with high lidar ratios (i.e., pollution and/or smoke particles) and (iv) small particles with medium lidar ratios (i.e., clean continental particles). Generally, desert dust layers have optical properties that are considerably different from the other types. Their big size leads to low Ångström exponent values and the reported lidar ratio at 355nm ranges from 47 to 58 sr for Thessaloniki (Siomos et. al., 2018). PollutedSmoke particles are highly absorbing particles, with high lidar ratio values. CleanContinental categorization is not completely straightforward, because the continental particles can be attributed to different subcategories (i.e., local, continental polluted or mixtures). In general, the CleanContinental cases are typically elevated layers, i.e. layers not related to the local atmospheric boundary layer where the pollution and anthropogenic contribution would mean more absorbing particles and therefore labeled as PollutedSmoke aerosol.

## 4 Application of the two automatic algorithms to EARLINET data - case studies

The application of the two automated aerosol typing algorithms on aerosol classification for three cases is presented and discussed in detail. A Saharan dust outbreak, a biomass burning case and a complex aerosol structure scene over Thessaloniki station are presented below. The typing results are compared against manually typed profiles that were characterized using satellite data and model simulations. Specifically, in order to identify the source of aerosol particles, backward trajectories are calculated using the Hybrid Single-Particle Lagrangian Integrated Trajectory model (HYSPLIT; Stein et al., 2015). Additionally, the BSC-DREAM8b model (e.g., Basart et al., 2012b) was used to verify the presence of Saharan dust. Finally, fire spots from the space-based MODIS sensor (https://firms.modaps.eosdis.nasa.gov/map/) product FIRMS (Fire Information for Resource Management System; Giglio et al., 2016) were used to confirm the presence of smoke over Thessaloniki.

## 4.1 Dust Case

The first case refers to the occurrence of a dust plume over the station of Thessaloniki on 20th of May 2013. The retrieved profiles of the particle backscatter and extinction coefficient, lidar ratios, and Ångström exponents are shown in Fig. 1 (a-d). The measurement is characterized by two particle layers, the first one between 0.7 and 1.6km and the second one between 2.0 and 2.7km. The dust presence was confirmed by the 4-days backward trajectories arriving at Thessaloniki on 20th of May 2013 (Fig. 1e). The trajectories indicated an event of transported Saharan dust. Additionally, the BSC-DREAM8b model is used to verify the presence of Saharan dust. The aerosol main layers and NATALI typing results are presented in Fig. 1f. For each aerosol layer (i.e., Layer 1: 0.7-1.6 km and Layer 2: 2-2.7 km), the mean optical properties are calculated using NATALI. The lidar ratio value for Layer 1 is 45±3 sr for 355nm and 43±1.1 sr for 532nm and for Layer 2 is 45±1.4 sr for 355nm and 41±1.3 sr for 532nm correspondingly. The ratio of the lidar ratios values ($LR_{532}/LR_{355}$) for Layer 1 is 0.94±0.083 and for Layer 2 is 0.92±0.13. The mean $AE_{355/532}$ is 0.33 ± 0.06 for Layer 1 and 0.38±0.07 for Layer 2 respectively. The mean $BAE_{355/532}$, mean $BAE_{532/1064}$ and mean $BAE_{355/1064}$ are 0.24±0.06, 0.28±0.04 and 0.32±0.22 for the first layer, while the values for the Layer 2 are 0.13±0.07, 0.06±0.04 and 0.05±0.1 respectively. The stability of the lidar ratio and Ångström exponent values could be considered as an indicator of homogeneity and small variability of the aerosol type within the layer. The retrieved values indicate the presence of coarse particles and are in agreement with the typical dust values observed over Thessaloniki (Siomos et al., 2018). Also, Müller et al., (2007) reported values of mean backscatter-related Ångström exponent $BAE_{355/1064}$ of 0.2±0.2 for desert dust events. NATALI typing output indicates dust layers (Fig. 1f). The EARLINET Mahalanobis distance-based typing algorithm also classifies Layers 1 and 2, as dust cases (Mahalanobis distance is minimum), and Mahalanobis probability also shows good predictive performance (68% Layer 1 and 44.6% Layer 2). (Fig. 1(g-h)). The plotted probabilities are assigned to each calculated distance and are different from the normalized ones that used as a screening criteria (and add up to 100).

## 4.2 Biomass burning case

The second case is a biomass burning episode that occurred on 2nd September 2013. The 3-day backward trajectories from HYSPLIT in conjunction with fire spots from MODIS satellite product FIRMS indicate the biomass burning episode, transported from central Europe in the region of Thessaloniki. The optical profiles and the layers of aerosols are shown in Figure 2 (a-d). The measurement is characterized by three particle layers: the first one is between 0.98km and 1.2km (Layer 1), the second one between 1.7 and 2.6km (Layer 2) and the third one between 2.9 and 3.5 km (Layer 3). The upper and lower boundary of each layer is marked with lines. NATALI typing is presented in Fig. 2f. The mean values of all optical parameters for each detected layer are calculated. The mean Lidar ratio is calculated 69±4 for 355nm and 70±4 for 532nm (Layer 1), for Layer 2 is 68±2 sr for 355nm and 69±2 sr for 532nm and Layer 3 has values of 69±2.3 sr for 355nm and 75±2.5 sr for 532nm. The ratio of the lidar ratios values ($LR_{532}/LR_{355}$) for Layer 1 is 1.02±0.004, for Layer 2 is 1.02±0.007 and for Layer 3 is 1.09±0.048. Mean Ångström (AE) exponent for each layer is estimated at 1.99±0.13 (Layer 1), 1.84±0.06 (Layer 2) and 1.61±0.08 (Layer 3). These values are consistent with previous reported ones, e.g., Baars et al. (2012) reported values of

the extinction Ångström exponent for smoke of 1.17 ± 0.44. The mean backscatter-related Ångström exponent (BAE) at 355-532nm is 2.03±0.13 (Layer 1), 1.88±0.06 (Layer 2) and 1.80±0.08 (Layer 3), while the mean backscatter-related Ångström exponent (BAE) at 532-1064nm is 1.43±0.07 (Layer 1), 1.08±0.04 (Layer 2) and 1.03±0.05 (Layer 3), and mean backscatter-related Ångström exponent (BAE) at 355-1064nm is 1.63±0.07 (Layer 1), 1.36±0.04 (Layer 2) and 1.31±0.03 (Layer 3),

respectively. These values are in accordance with the typical biomass burning values observed over Thessaloniki. Giannakaki et al. (2010) reported an annual mean lidar ratio at 355nm of 69 ± 17 sr and a mean BAE at 355-532nm of 1.7 ± 0.7, while Siomos et al. (2018a) found lidar ratio at 355nm ranging from 51 to 73 sr for biomass burning events. NATALI aerosol output (Fig. 2f) and the EARLINET Mahalanobis distance-based typing algorithm values (distance and probability) also confirm the presence of smoke layers (Fig. 2(g-k)). The plotted probabilities are assigned to each calculated distance and are different from

the normalized ones that are used as a screening criteria (and add up to 100).

## 4.3 Mixed case

The third case is a case with complex aerosol structure that occurred on 21st August 2014. This case offers the opportunity to check the reliability of the algorithms in conditions where different aerosol types in different layers exist. The 5-day backward simulation from HYSPLIT indicates the pattern of the origin of air masses before reaching the study area as presented in

Figure 3e. The motion of the particles is in a southwest direction from a dust source towards Thessaloniki. So, the path of the air masses arriving over Thessaloniki suggests a mixture of dust (at height of about 3 km) and continental and marine particles. The optical profiles and the layers of aerosols are shown in Figure 3 (a-d). The measurement is characterized by four particle layers: the first one is between 0.9km and 1.10km (Layer 1), the second one between 1.5 and 2.1km (Layer 2), the third one between 2.5 and 3.8 km (Layer 3) and the forth one between 4.47 and 5.52km. The upper and lower boundary of each

layer is marked with lines. NATALI typing is presented in Fig. 3f. The mean values of all optical parameters for each detected layer are calculated. The mean Lidar ratio is calculated 56±3.17 for 355nm and 37±2.09 for 532nm (Layer 1), for Layer 2 is 57±1.75 sr for 355nm and 44±1.35 sr for 532nm, Layer 3 has values of 43±0.91 sr for 355nm and 42±0.68 sr for 532nm and Layer 4 has values of 29±0.77 sr for 355nm and 30±0.91 sr for 532nm. The ratio of the lidar ratios values ($LR_{532}/LR_{355}$) for Layer 1 is 0.66±0.08, for Layer 2 is 0.77±0.05, for Layer 3 is 0.97±0.067 and for Layer 4 is 1.03±0.1. Mean Ångström

exponent (AE) for each layer is estimated at 2±0.14 (Layer 1), 0.86±0.08 (Layer 2), 0.85±0.05 (Layer 3) and 1.72±0.07 (Layer 4). The mean backscatter-related Ångström exponent (BAE) at 355-532nm is 0.98±0.14 (Layer 1), 0.19±0.08 (Layer 2), 0.10±0.05 (Layer 3) and 1.21±0.06 (Layer 4). The mean backscatter-related Ångström exponent (BAE) at 532-1064nm is 0.77±0.08 (Layer 1), 0.68±0.04 (Layer 2), 0.58±0.03 (Layer 3) and 1.9±0.03 (Layer 4). The mean backscatter-related Ångström exponent (BAE) at 355-1064nm is 0.78±0.07 (Layer 1), 0.49±0.04 (Layer 2), 0.39±0.03 (Layer 3) and 1.64±0.03

(Layer 4), respectively. NATALI aerosol output is presented in Fig. 3f. NATALI classifies as Unknown the Layer 1 and Layer 2, as the ANNs could not identify any aerosol type (no ANNs passed the threshold criteria), whilst classifies Layer 3 as Dust and Layer 4 as Maritime. The EARLINET Mahalanobis distance-based typing probability values are presented in Fig. 3 (g-j). The Layer 1 is classified as CleanContinental, the Layer 2 and the Layer 3 as Dust and the Layer 4 as CleanContinental. Both typing techniques classify Layer 3 as Dust. The values of the optical parameter indicate the presence of coarse particles and are

in agreement with the typical dust values observed over Thessaloniki (Siomos et al., 2018). Layer 4 is typed differently. This layer is recognized by NATALI as Maritime one (lower Lidar Ratio values), while in the EARLINET Mahalanobis distance-based typing method is attributed as signature for CleanContinental. Layer 1 and Layer 2 are typed as Unknown layers from NATALI. However, Layer 1 is recognised as CleanContinental by EARLINET Mahalanobis distance-based typing method,

with mean backscatter-related Ångström exponent and Lidar Ratio values, representative of the continental particles, allowed in the EARLINET Mahalanobis distance-based typing method (Table 2). Given that, the two algorithms take into consideration different combination of the intensive optical properties (i.e, NATALI uses AE), this is also a reason that differentiate the output retrieval. The layer high Ångström exponent of Layer 1, considered in the NATALI typing algorithm, maybe is the reason that ANNs answers are filtered out. Same for Layer 2, NATALI return the layer as Unknown, due to the low confidence level of the

ANN outputs. However, the EARLINET Mahalanobis distance-based typing method classifies the Layer as Dust, based on the values of the optical parameters used for typing (Table 2). This example demonstrates that type assignment is quite challenging and not always possible in ambiguous atmospheric scenes.

## 4.4 Automated aerosol classification using NATALI and EARLINET Mahalanobis distance-based typing algorithm

The complete Thessaloniki multiwavelength Raman lidar dataset for the 2012-2015 period was analyzed in terms of aerosol

typing with the NATALI and the EARLINET Mahalanobis distance-based typing algorithm using the automated methodologies reported in Sect. 3.1 and 3.2. For the 54 cases, 120 layers are identified by the layer identification module of NATALI, out of which 10 layers are rejected due to the fact that they were below 1 km (below overlap region).

NATALI classified 82 of the total 110 layers (75% of the total number of cases) and 28 were flagged as Unknown layers. These missed cases are based mostly on retrievals with large uncertainties of the optical parameters and the calculated uncer-

tainties and rely on the minimum acceptance confidence that prohibited ANNs from returning an aerosol type classification. However, these 'unknowns' layers provided type results (93%), when the minimum acceptance confidence was set to 70%. However, in this study we investigate the application of NATALI algorithm holding the retrievals that pass a higher percentage of quality checks. The EARLINET Mahalanobis distance-based typing algorithm classified 98 layers (89%) and 12 layers were not assigned to any cluster (the Mahalanobis distance values larger than 4), showing a higher identification rate. However, this

rate also depends on the screening criteria. When we set the second of the screening criteria (i.e., the normalized probability of the aerosol class) to be higher than 50%, the number of untyped layers increased (22 untyped cases, 80% typing rate). Overall, the identification rate can be changed regarding the needs of the user and the selection of the appropriate threshold criteria. In addition, we examined the ability of the both methods to provide typing results, relative to the derived layer optical depth (Aerosol Optical Depth - AOD) values. This study showed that the mean layer AOD of the typed layers has similar value for

both typing techniques (0.1076 ± 0.0898 (SD) for NATALI and 0.11166 ± 0.0928 (SD) for the EARLINET Mahalanobis distance typing algorithm). However, the EARLINET Mahalanobis distance algorithm seems more sensitive to lower values (not typed cases show a mean layer AOD of 0.0298 ± 0.0578 for the EARLINET Mahalanobis distance algorithm, in contrast to the 0.06899 ± 0.0606 for NATALI). These low AOD values are associated with higher uncertainties on the intensive properties.

Figure 4 (a,c) presents the percentages of the total typed layers into the defined aerosol classes for both automated algorithms. Also, the untyped layers from the EARLINET Mahalanobis distance-based typing algorithm that were typed by NATALI and visa versa are presented in Fig. 4 (b,d). The EARLINET Mahalanobis distance-based typing algorithm, classified 27 of the 28 untyped layers by NATALI, as follows: 6 layers as Dust (21.4%), 10 as PollutedSmoke (35.7%), 10 as CleanContinental (35.7%) and only 1 layer as Maritime (3.6%). Here again, the EARLINET Mahalanobis distance-based typing algorithm, indicated PollutedSmoke and CleanContinental category as the main sources of the aerosol layers observed, while Maritime category is the less pronounced one. On the other hand, NATALI classified 11 of the 12 untyped layers from the EARLINET Mahalanobis distance-based typing algorithm, as follows: 2 layers were typed as Dust (16.7%), 6 as PollutedSmoke (50%) and 3 layers as Maritime (25%). In general, NATALI software seems more sensitive to the Maritime category. In particular, the agreement is reasonably close for the desert dust cases (10% and 17% for NATALI and EARLINET Mahalanobis distance-based typing algorithm, respectively), nevertheless, it becomes evident that the particle linear depolarization ratio could increase the ability for correctly predicting dust particles (Fig. 4). The agreement is remarkable for the PollutedSmoke cases, with occurrence ratio of 43% and 47% for NATALI and EARLINET Mahalanobis distance-based typing algorithm, respectively, whilst the CleanContinental cases showed a difference with 18% and 24% for NATALI and EARLINET Mahalanobis distance-based typing algorithm, respectively. Also, difference is observed for the Maritime category, where one case is attributed to this category by NATALI.

The differences observed in Fig. 4, primarily, can be described by the different definition of the aerosol classes for the two typing methods. It is worth noting that the aerosol model for the NATALI typing method is trained based on synthetic data. The comparison between the aerosol model values used for the training and the reported literature values showed differences (Nicolae et al., 2018), as follows: Synthetic values found lower than those observed for continental-rural (AE 350/550), continental-polluted (CR 500/1000), and dust (CR 500/1000) types and synthetic values are greater for continental-rural (LR350) and volcanic (DEP 550) types. The reasons for these differences could be related on one side to uncertainty or variability issues of the measured values and on the other side to the aerosol model limitations e.g. due to spheroidal model and mono-modal log-normal distribution (Nicolae et al., 2018). On the other hand, for the EARLINET Mahalanobis distance-based typing method the aerosol reference class is defined using an ensemble of high quality EARLINET observations. However, different measurement locations may be a reason for mismatches in the attributed aerosol category. Moreover, the general different approaches of the two methods may have an impact on the aerosol typing, even though both classifications schemes have been trained using supervised learning techniques. Additional to the ability of the two automated algorithms to attribute the correct type to the aerosol layers, also the uncertainties of the input measurements have to be taken into account. Both algorithms are unable to return aerosol type for not well calibrated input data, i.e., when one or more intensive optical properties have large uncertainties.

In what follows, a comparison is made to the 71 common layers, identified by both algorithms. Figure 5 presents the number of detected layers per typing class attributed by each algorithm for the whole study period. The two algorithms attributed the same aerosol type to 56 layers, showing an agreement of 78.8% of the typed cases. The typing procedures show the predominance of the PollutedSmoke category for Thessaloniki, followed by the CleanContinental category. This result is

consistent with the CALIPSO scheme at global level and with results about Thessaloniki site characterization done by Siomos et. al (2018b). In their paper Siomos et al. (2018b), used data from a double monochromator Brewer spectrophotometer and a sunphotometer in order to classify aerosol cases during the period 2007-2017 in Thessaloniki: they found that the Water Soluble category (which can be related to the Clean Continental consisting of water soluble particles) correspond to 29.1% of the cases,

which is in fair agreement with results reported in Fig. 5. Dust results in a smaller amount of the observed layers, according to the retrievals of both algorithms. Finally, although Thessaloniki is a coastal site, the maritime layers are rare, presumably, due to the mixing with other aerosol types. These results are in agreement with the findings of Siomos et al. (2018).

Although, each automated classification algorithm has important differences acknowledged above, the comparison showed an overall good agreement for the four defined aerosol classes. The convergence of the two different methods on the same type

can be regarded as a signature of reliability. An almost perfect score was found for the PollutedSmoke category, following by the Dust one, given that the dust class is well defined for both typing schemes, as the physical properties of dust particles differentiate from the other three classes. Considering the mismatches in the CleanContinental category, good agreement is found. By contrast, the maritime category is defined in a different way for the two automated algorithms. The EARLINET Mahalanobis distance-based typing algorithm considers maritime layers mixed with other aerosol types, whereas for the NATALI

the mixing is negligible and the aerosol type refers to pure maritime aerosol. The absence of measurements for such kind of particle also did not allowed a direct assessment of the pure marine particle synthetic data into NATALI algorithm itself. The case typed as Maritime by NATALI was identified as CleanContinental: this is because of the different lidar ratio at 532nm and backscatter Ångström related values allowed in the NATALI scheme which are recognized by the EARLINET Mahalanobis distance-based typing method as signature for CleanContinental types (Table 2).

Discussion of the mismatch cases is extremely important for understanding the limitations of the assumptions behind the two schemes and provide a useful data set to investigate the causes, which will help to potentially improve the typing methods. In what follows, the NATALI aerosol model dataset is ingested into the EARLINET Mahalanobis distance-based typing algorithm with the aim of understanding the differences in the aerosol definition. 259407 out of 424236 instances (61%) were typed by the EARLINET Mahalanobis distance-based typing algorithm. The not-typed instances were discarded by the algorithm owing

to a distance-threshold filter and a distance-similarity filter (see Sect. 3.2). The results of the application of the EARLINET Mahalanobis distance-based method to the modeled reference classes of NATALI are presented in Fig. 6. Each percentage is defined here as follows: The denominator in each column is the number of NATALI samples for a given type, and the numerator is the number of the EARLINET Mahalanobis distance-based classifications for the same sample.

Fair agreement is found for the NATALI synthetic datasets of Dust and PollutedSmoke categories. In particular, for Dust

reached agreement of 83.7% (100166 typed over 119650). This result is very good considering that up to 30% of the aerosol class contains mixtures. For the PollutedSmoke, the majority of the synthetic data, corresponding to 48180 instances from a total typed of 82041 (58.7%) is classified as PollutedSmoke. For the CleanContinental type, the agreement is poor (15%), as only 7605 over 52058 were correctly classified and the majority of the instances (36552) is typed as Dust (70%). This Dust-CleanContinental mismatch is confirmed also by the 11.8% of the NATALI reference dust cases that are typed as Clean

Continental by the EARLINET Mahalanobis distance-based typing method. Furthermore, 7901 of the instances (15%) are

classified as PollutedSmoke. Finally, the agreement for Maritime is 23% of the cases (1319 from a total of 5658 cases), indicating the different class definitions as well as the big spread of the NATALI maritime model that causes the EARLINET Mahalanobis distance-based typing algorithm to misclassify those data.

Despite the important differences found on the set-up of the two methods, a very good agreement is achieved when the algorithms are applied on the EARLINET Thessaloniki dataset as shown in Fig. 7. For PollutedSmoke and CleanContinental the accuracy reached 88% (the common number of typed layers as PollutedSmoke from both algorithms to the total layers typed as PollutedSmoke from NATALI) and 65% (the common number of typed layers as CleanContinental from both algorithms to the total layers typed as CleanContinental from NATALI) respectively. A good performance is also found for the Dust category (78%). It is obvious, that the additional information of the particle depolarization ratio will further improve the performance of the automated typing algorithms in distinguishing dust particles from other aerosol types. A not satisfactory agreement is observed for the Maritime, which is the aerosol type less encountered over Thessaloniki. Overall, there were 15 cases (on Thessaloniki dataset) that the two methods provided different typing results. There are seven cases typed as CleanContinental aerosol by NATALI and PollutedSmoke or CleanContinental by EMD, five cases typed as PollutedSmoke by NATALI and Dust by EMD, 2 cases typed as Dust by NATALI and CleanContinental by EMD and one case type as Maritime by NATALI and CleanContinental by EMD. These mismatches are illustrated in Figure 7. In order to understand these differences we highlight some critical issues relevant to the type definitions of the two methods, based on Nicolae et al., (2018) and Papagiannopoulos et al., (2018):

CleanContinental: The contribution of the Soot component in the chemical composition of the cleancontinental category, allows higher lidar ratio values at 532nm (52–53sr) in the NATALI scheme. Consequently, layers recognized by NATALI as CleanContinental ones, in the EARLINET Mahalanobis distance-based typing method are attributed as signature for PollutedSmoke or Dust (Figure 7).

Marine: As observations of pure maritime particles are quite scarce within EARLINET and, generally, when these particles are observed their characteristics are far from pristine, the Maritime category for the EARLINET Mahalanobis distance typing algorithm, corresponds to mixed maritime layers. This is different from the pure maritime category that NATALI identifies. Lower lidar ratio values for 532nm (19–25sr) are defined in the NATALI software for the identification of marine layers, in contrast to the higher ones (16-32sr) allowed in the EARLINET Mahalanobis distance typing algorithm.

Dust: Higher values of Lidar ratio at 532nm are allowed for EARLINET Mahalanobis distance typing algorithm in identifying dust particles (41–63sr), considering all dust-like aerosol types as one category, while NATALI allows values corresponding to more pure cases (44–49sr). Therefore, a number of Dust recognition are attributed to PollutedSmoke particles (6 cases) or CleanContinental particles (1 case) in the NATALI output.

PollutedSmoke: An almost perfect score was found for the PollutedSmoke category. This can be attributed to the similar reference values attributed by both typing algorithms. The values of Lidar ratio at 532nm allowed for NATALI are in the range of 62-92sr (ContinentalPolluted and Smoke) and the ones allowed in the EARLINET Mahalanobis distance typing algorithm are in the range of 50–97sr.

As discussed above, the ability and performance of both typing methods strongly depend on the differences on the reference aerosol classes definition and the selection of the range thresholds applied to the intensive optical properties. The latter follow a different approach in the typing schemes (are linked with uncertainties in the NATALI algorithm and with the normalized probability based on the calculated distanses in the Mahalanobis distance typing algorithm).

## 5   Summary and Conclusions

In this study, two automated typing methods are used to obtain the dominant aerosol type using optical properties, measured by a multiwavelength Raman lidar over Thessaloniki. The application of the automatic classification methods in three case studies shows consistent results when compared against manually classified EARLINET data. Additionally, all Raman measurements for the period 2012 to 2015 are analyzed and a coarse aerosol classification analysis is made. The classification analysis covers the four major aerosol classes, representing conditions in the area of Thessaloniki, i.e, Dust, Maritime, PollutedSmoke and CleanContinental.

Both classification schemes indicate PollutedSmoke and CleanContinental category as the main sources of the aerosol layers observed in 3 years of measurements over Thessaloniki, while Dust accounts for about 10-17% of the cases and Maritime particles are rarely observed. Remarkable agreement is found in the aerosol typing for the PollutedSmoke category (88%), followed by Dust (78%), whilst fair agreement is observed for the CleanContinental category (65%). Maritime is the worst-performing category, underlining the need for further investigation on the identification and validation of optical property thresholds to identify sea salt aerosol. The convergence of the two different methods on the same type can be regarded as a signature of reliability, while differences can be ascribed to the different aerosol type definition. Moreover, despite the differences in the optical property thresholds allowed in each typing scheme, many of these thresholds do overlap (especially in the low resolution mode). This might result to allow typing into multiple categories. In general, it is worth mentioning that, as demonstrated in Nicolae et al. (2018) and Papagiannopoulos et al. (2018), the availability of the particle linear depolarization ratio improves the predictive accuracy of both methods. Its availability could enhance the strength of correct predictions, especially for the dust particles recognition and could lead to the increase of the number of detected types (high resolution typing) for both algorithms. It should also, be mentioned that the success rate of the aerosol classification schemes (and thus their comparability) is largely dependent on the uncertainties of the optical properties and the threshold criteria defined. Future analysis comparing the two typing schemes on the high resolution mode, as well as further application on other lidar stations with more complex locations and topography, is ongoing work for determining the validity of the comparison made in this study.

Overall, we conclude that the application of these typing techniques could contribute to the improvement of the model treatment and to a better exploitation of the present and future data from satellite remote sensing for aerosol classification schemes. Assessments about the adaptability of the presented aerosol type classification schemes to future satellite missions are ongoing work. Considering that up to now, there is not a well established method to provide correct results under every

conditions, the systematic and full comparison of different methods is the only way to have a better insight about the possibility of inferring the aerosol source from the observed optical properties. Additionally, the modularity of the EARLINET Single Calculus Chain tool offers the possibility to implement aerosol typing procedures for automatically providing the aerosol typing information. A comparison between the two existing typing schemes that make use of the aerosol optical properties could be rather useful, as it will allow from one side the optimization of aerosol typing models and on the other side the

definition of the aerosol types used and the selection of the reference dataset.

*Code availability.* The lidar data used in this study are available at http://data.earlinet.org. NATALI algorithm is publicly available at http://natali.inoe.ro/resources.html/software. The EARLINET distance-base typing algorithm and related reference datasets will be soon available at www.earlinet.org .

*Author contributions.* D. Balis, N. Siomos, and KA. Voudouri performed and processed lidar measurments during the period 2012–2015.

N. Papagiannopoulos developed the EARLINET Mahalanobis distance-based typing algorithm. D. Nicolae developed the NATALI aerosol typing algorithm. K. Michailidis carried out the processing of lidar measurements with the EARLINET Mahalanobis distance-based typing algorithm. KA. Voudouri carried out the processing of lidar measurements with NATALI algorithm and prepared the figures. L. Mona reviewed parts of the comparison of the two algorithms. D. Balis is the PI of the lidar station and directed the preparation of the manuscript. KA. Voudouri prepared the manuscript with contributions from all co-authors.

*Competing interests.* The authors declare that they have no conflict of interest.

*Acknowledgements.* The research leading to these results has received funding from the the European Union's Horizon 2020 Research and Innovation programme for Societal challenges - smart, green and integrated transport, under grant agreement no. 723986 (project EUNADICS –AV – European Natural Disaster Coordination and Information System for Aviation). Voudouri K.A acknowledges the support of the General Secretariat for Research and Technology (GSRT) and the Hellenic Foundation for Research and Innovation (HFRI), Scholarship code: 95041.

The financial support for EARLINET in the ACTRIS Research Infrastructure Project by the European Union's Horizon 2020 research and innovation program under grant agreement no. 654169 in the Seventh Framework Programme (FP7/2007–2013) and the ECARS project (East European Centre for Atmospheric Remote Sensing), under grant agreement n. 602014, ACTRIS-2 project, are gratefully acknowledged.

Further financial support is provided by the Romanian Ministry of Research and Innovation throughout the Core National Program no. 33N/PN2018.

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

**Table 1.** Nomenclature of the aerosol types used in the High and Low resolution (without depolarization) mode for both automatic typing techniques and the correspondence with the ones used in this study.

| NATALI | | EARLINET Mahalanobis | | This Study |
|---|---|---|---|---|
| High Resolution | Low Resolution | High Resolution | Low Resolution | Low Resolution |
| Continental | Continental | CleanContinental | Continental | CleanContinental |
| ContinentaPolluted, ContinentalSmoke, Smoke, MixedSmoke/CC | Continental polluted, Smoke, Continental Smoke | Polluted Continental, Smoke | PollutedSmoke | PollutedSmoke |
| Dust, Volcanic, Dust-Polluted, MixedDust/CC, ContinentalDust, Mineral Mixtures/Volcanic | Dust | Dust, Volcanic, Mixed Dust, Polluted Dust | Dust, Polluted Dust | Dust |
| Marine, CoastalPolluted/CC, Coastal/CC | Marine | Mixed Marine | Mixed Marine | Maritime |

**Table 2.** Mean aerosol optical properties of the reference aerosol types used on the two automated algorithms.

| NATALI | $AE_{350/550}$ | $LR_{350}$ | $LR_{550}$ | $CR_{350/550}$ | $CR_{550/1000}$ |
|---|---|---|---|---|---|
| Dust (D) | 0.88–0.92 | 43–46 | 44–49 | 1.51–1.55 | 1.1–1.14 |
| ContinentalPolluted (CPolluted) | 1.17–1.34 | 55–75 | 62–74 | 1.34–2.29 | 1.33–1.65 |
| Smoke (S) | 1.15–1.31 | 56–72 | 81–92 | 1.90–2.59 | 1.52–1.61 |
| Continental (CC) | 1.17–1.29 | 43–54 | 52–53 | 1.56–2.07 | 1.37–1.85 |
| Maritime (M) | -0.26–0.21 | 13–32 | 19–25 | 0.77–1.35 | 0.7–2.91 |
| EARLINET Mahalanobis distance-based typing algorithm | | | $BAE_{355/1064}$ | $LR_{532}$ | $LR_{532}/LR_{355}$ |
| Dust (D) | | | 0.01–1.0 | 41–63 | 0.77–1.04 |
| PollutedSmoke (PS) | | | 1.0–1.9 | 50–97 | 0.87–1.6 |
| CleanContinental (CC) | | | 0.7–1.6 | 32–44 | 0.62–1.04 |
| Maritime (M) | | | 0.5–1.2 | 16–32 | 0.81–0.9 |

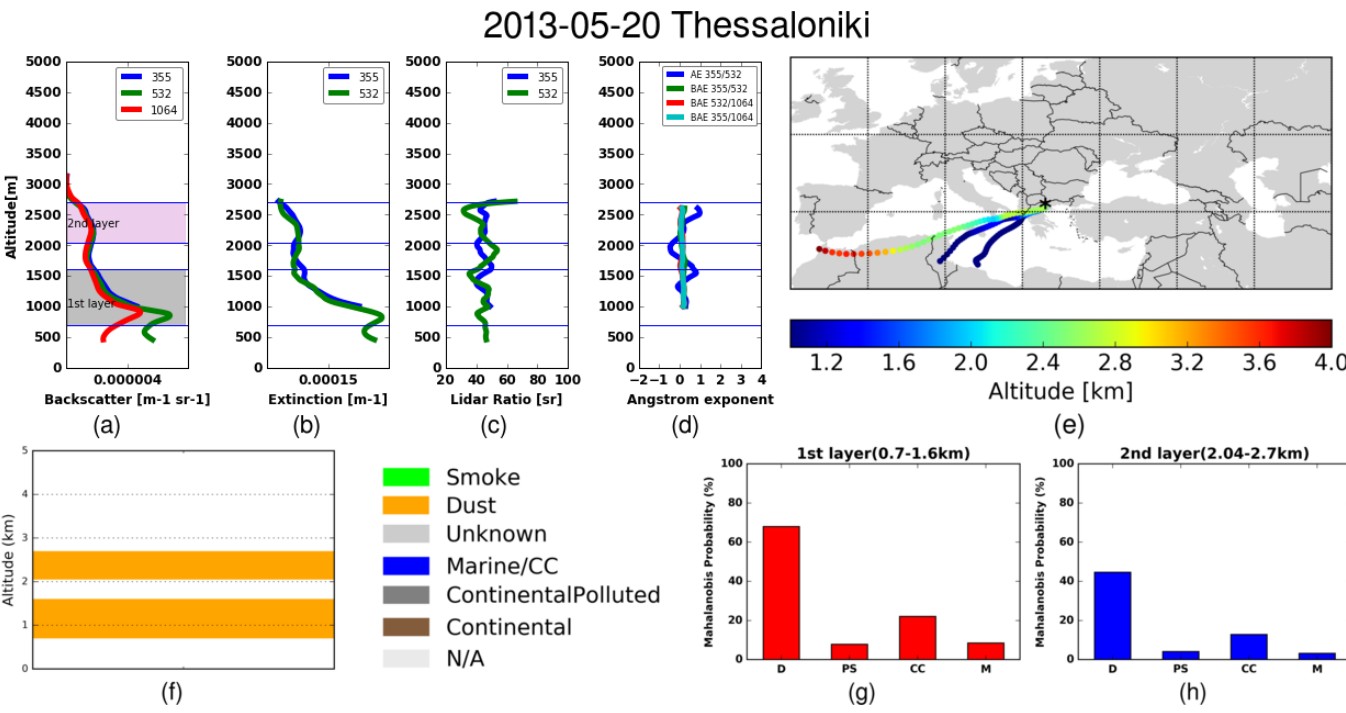

**Figure 1.** Vertical profiles of aerosol optical properties for Thessaloniki on 20th May 2013. (a) Backscatter coefficient at 355nm, 532nm and 1064nm, (b) extinction coefficient at 355 nm and 532nm, (c) Lidar Ratio, (d) Ångström exponent, (e) 4-days HYSPLIT backward trajectories arriving at Thessaloniki at each detected layer height, (f) aerosol main layers typed by NATALI and (g,h) EARLINET Mahalanobis distance-based typing algorithm probability for each detected layer (different from the normalized one).

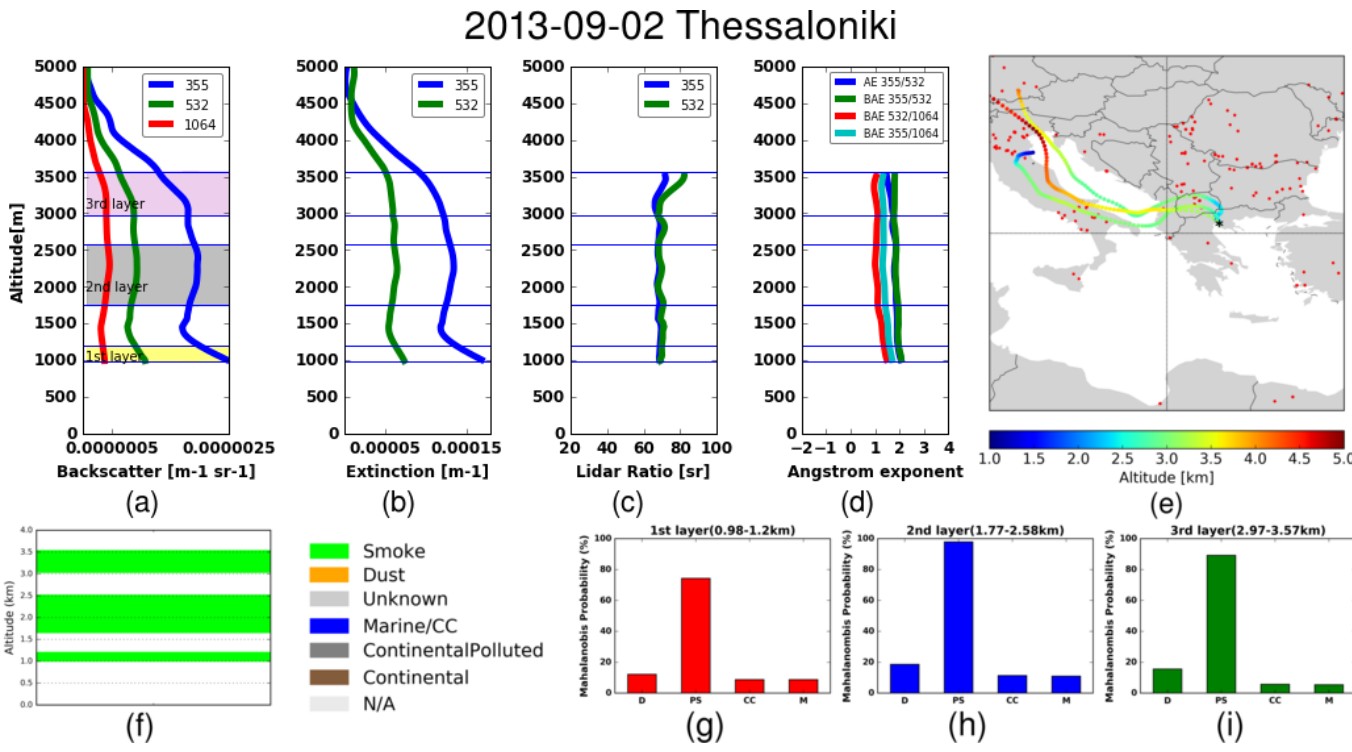

**Figure 2.** Vertical profiles of aerosol optical properties for Thessaloniki on 2nd of September 2013. (a) Backscatter coefficient profiles at 355nm, 532nm and 1064nm, (b) extinction coefficient profiles at 387nm and 607nm, (c) Lidar Ratio, (d) Ångström exponent, e) 3-days backward trajectories arriving at Thessaloniki at each detected layer height, (f) aerosol main layers typed by NATALI, and (g,h,i) EARLINET Mahalanobis distance-based typing algorithm probability for each detected layer (different from the normalized one).

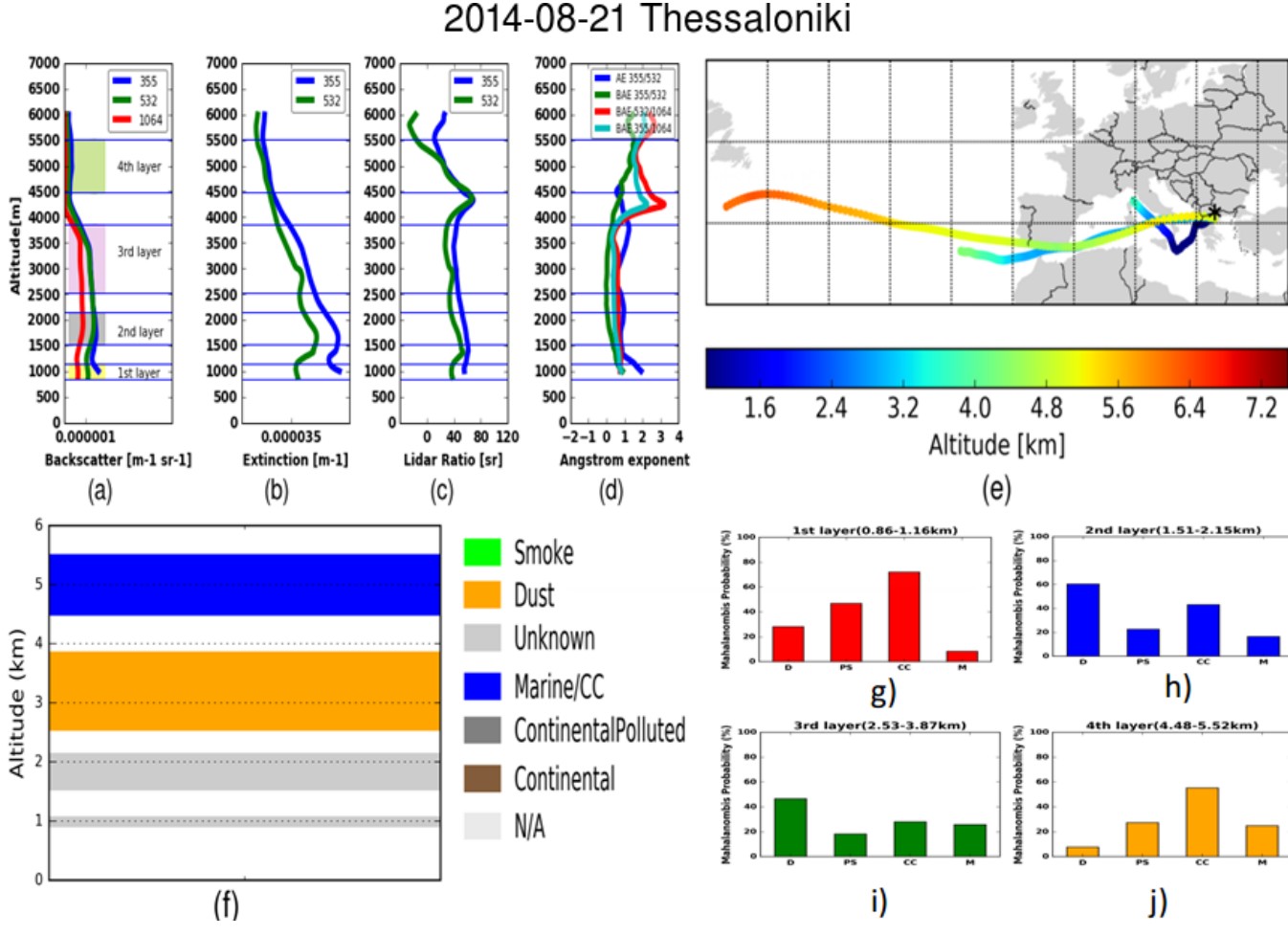

**Figure 3.** Vertical profiles of aerosol optical properties for Thessaloniki on 21st of August 2014. (a) Backscatter coefficient profiles at 355nm, 532nm and 1064nm, (b) extinction coefficient profiles at 387nm and 607nm, (c) Lidar Ratio, (d) Ångström exponent, e) 5-days backward trajectories arriving at Thessaloniki at each detected layer height, (f) aerosol main layers typed by NATALI, and (g,h,i,j) EARLINET Mahalanobis distance-based typing algorithm probability for each detected layer (different from the normalized one).

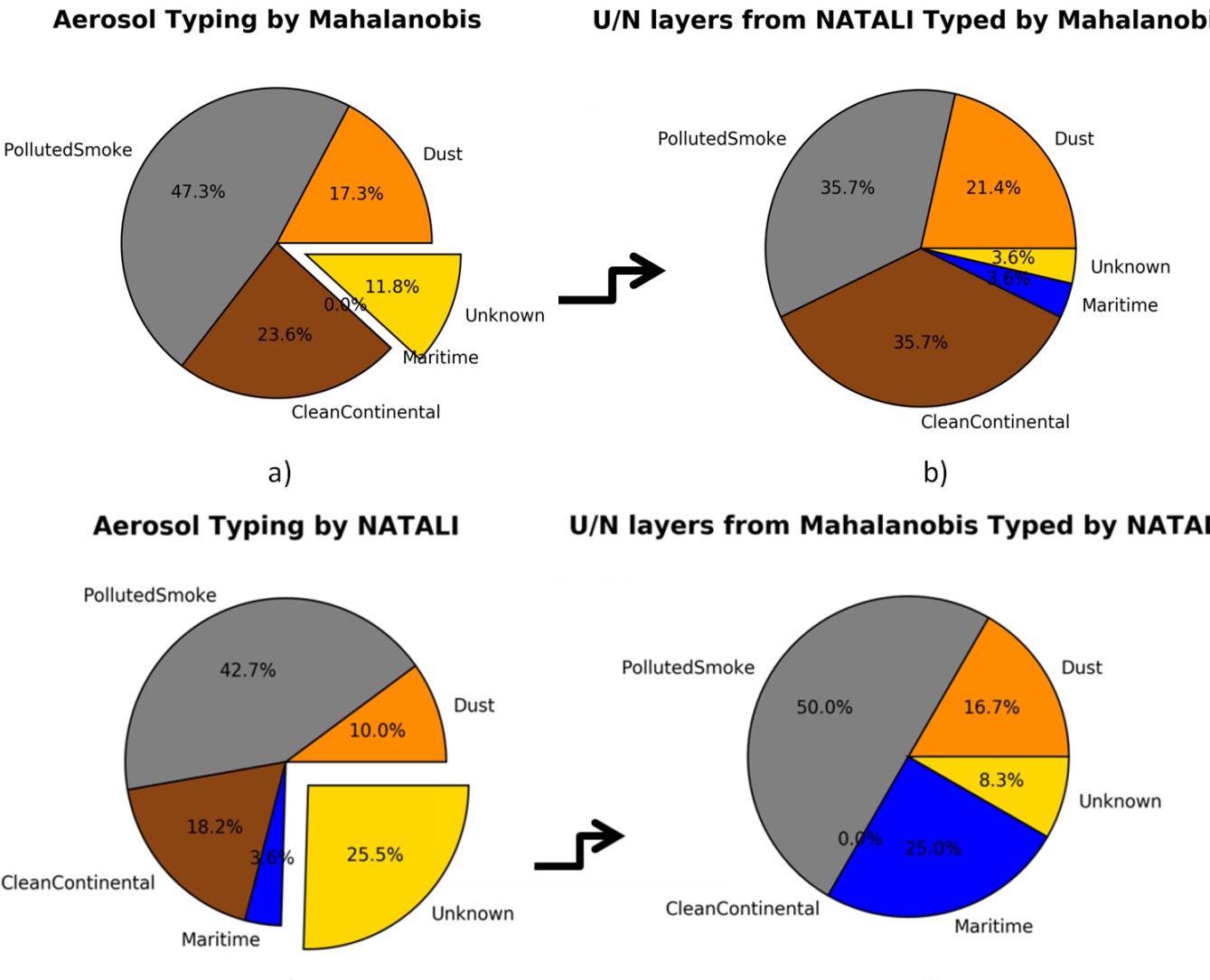

**Figure 4.** Percentages of typed and untyped layers for both automated algorithms.

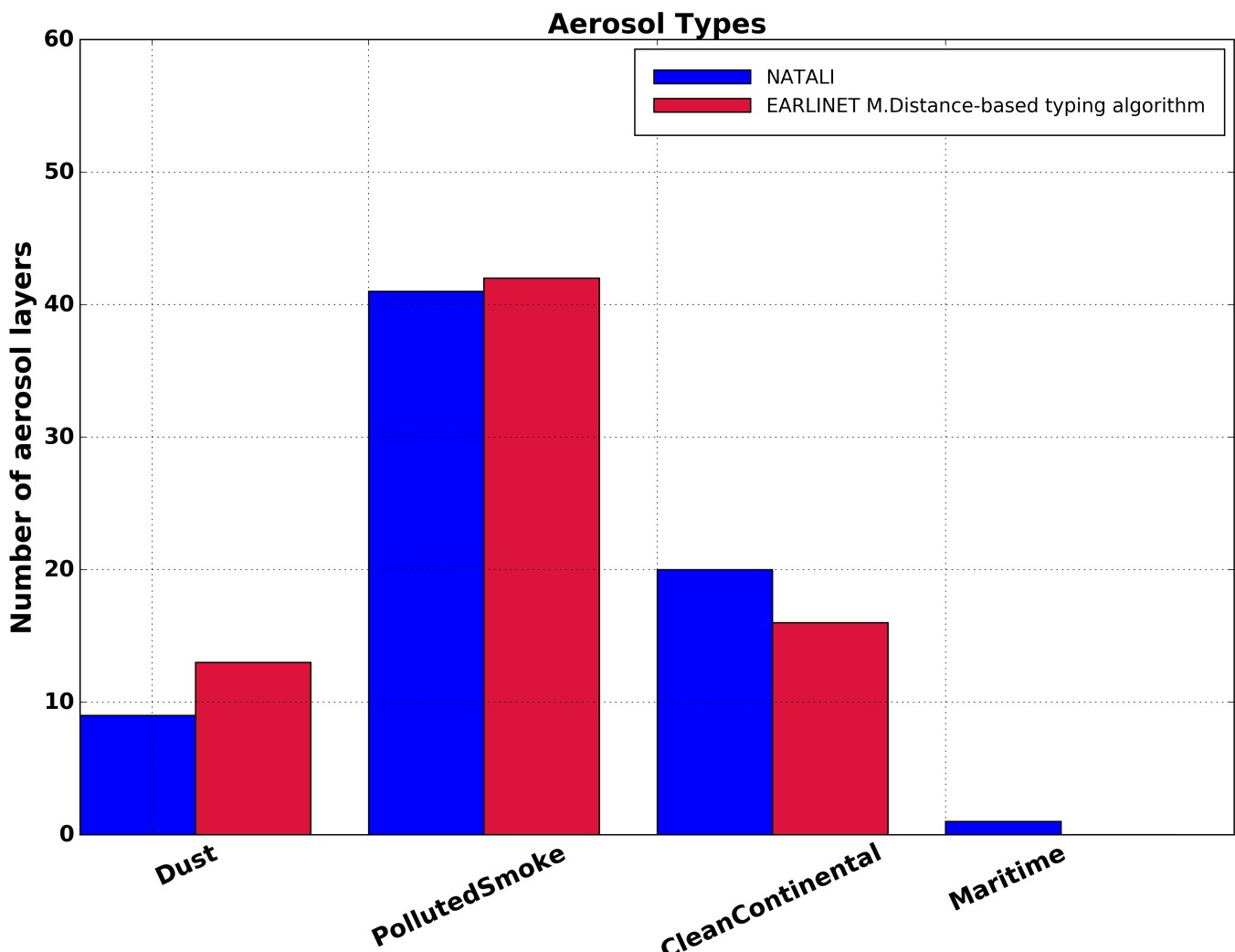

**Figure 5.** The number of detected layers form both algorithms, based on the 71 cases that both algorithms provided results.

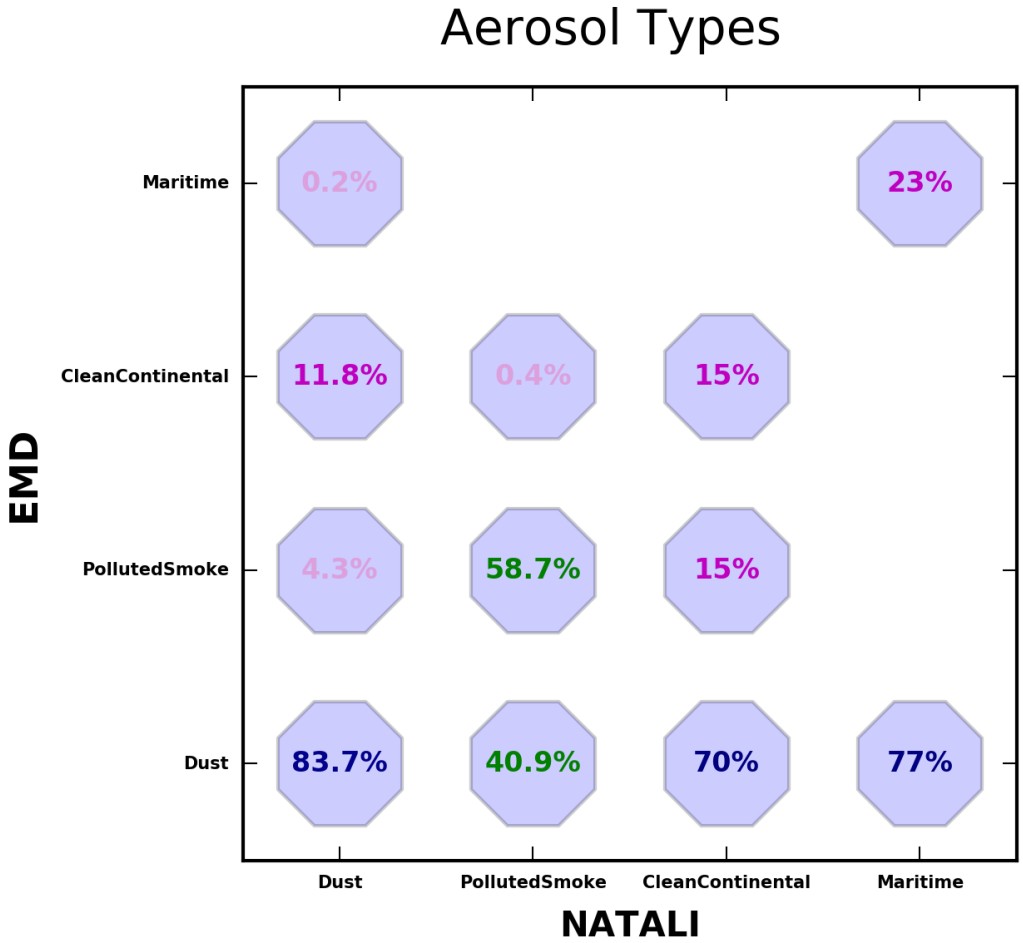

**Figure 6.** Typing score of each aerosol class derived by the EARLINET Mahalanobis distance-based typing algorithm (EMD), when applied to NATALI modeled data set (total of 259407 cases). The denominator in each column is the number of NATALI samples for a given type, and the numerator is the number of the EARLINET Mahalanobis distance-based classifications for the same sample.

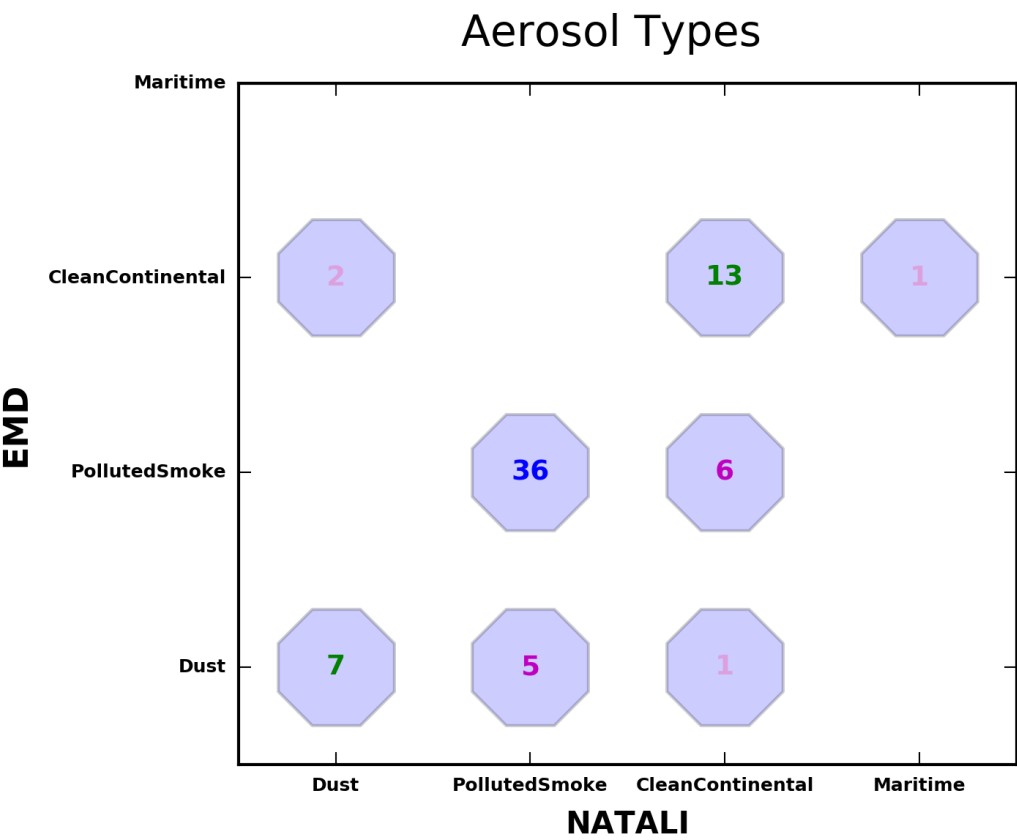

**Figure 7.** Number of cases per aerosol type derived by the two algorithms (EARLINET Mahalanobis distance-based typing algorithm - EMD and NATALI) for the period 2012-1015 (71 cases).