# Peer review of "Comparison of two automated aerosol typing methods and their application on an EARLINET station."

_Atmospheric Chemistry and Physics, 2018_

## Referee Comment (RC1) · Anonymous Referee #1 · 17 Jan 2019

**Review of "Comparison of two automated aerosol typing methods and their application on an EARLINET station" by K. A. Voudouri, N. Siomos, K. Michailidis, N. Papagiannopoulos, L. Mona, C. Cornacchia, D. Nicolae, and D. Balis**

As indicated by the title, this paper uses EARLINET $3\alpha + 2\beta$ lidar measurements to compare the performance of two different aerosol typing algorithms: the Papagiannopoulos (AKA Mahalanobis distance) algorithm and the NATALI algorithm from Nicolae et al.

For the Papagiannopoulos algorithm, aerosol classes are defined using an ensemble of high quality EARLINET measurements that has been manually partitioned by subject matter experts to ensure uniformity of the optical characteristics of all observations within each class. For the Nicolae/NATALI algorithm, aerosol classes are defined using "values obtained from a specially designed aerosol model with simulations from over 50,000 synthetic cases of [*different types of*] aerosols". Both classification schemes are trained using supervised learning techniques (although this point is not mentioned in the manuscript).

The lidar measurements used as algorithm inputs for the Papagiannopoulos algorithm are a) backscatter-related Ångström Exponent at 355nm and 1064nm ($AE_{355/1064}$), b) the lidar ratio at 532 nm ($LR_{532}$), and c) the ratio of the lidar ratios ($LR_{532}/LR_{355}$)

The lidar measurements used as algorithm inputs by the Nicolae/NATALI algorithm are not specified in this manuscript.

The high resolution variant of NATALI identifies 14 distinct aerosol types. The "low resolution without depolarization" variant used in this study recognizes five aerosol types: dust, smoke, continental polluted, continental, and maritime (page 6, line 15–16). For this study, the smoke and continental polluted types were combined into a single 'polluted smoke' type, so that only four types are recognized: dust, polluted smoke, continental, and maritime.

The standard Papagiannopoulos algorithm identifies 8 aerosol types: clean continental, polluted continental, pure dust, mixed dust (dust + maritime), polluted dust (dust + smoke and/or pollution), mixed maritime, smoke, and volcanic. For this study, these 8 types are combined to yield the same four types (page 7, line 1–2) that are recognized by the modified NATALI algorithm: dust (dust + volcanic + mixed dust + polluted dust), polluted smoke (smoke + polluted continental), clean continental, and maritime (mixed maritime).

Both algorithms were applied to the identical set of lidar measurements of 116 different aerosol layers. The NATALI algorithm successfully classified 80 of the 116 layers (~69%). The Papagiannopoulos algorithm successfully classified 114 of the 116 layers (~98%). The subsequent analyses are largely devoted to comparing classification results for the 80 cases in which both algorithms were successful.

The poor performance of the NATALI algorithm when applied to real-world lidar measurements suggests a disparity between the NATALI aerosol models and the EARLINET measurements. Similar poor performance (39% failure rate) was found in a simulation study in which the authors used the NATALI aerosol model to generate inputs for the Papagiannopoulos algorithm. Unfortunately, the authors do not investigate the possible causes or sources of these disparities.

In its present form, I cannot recommend this manuscript for publication. Most importantly, this study does not appear to address any specific science questions. Instead, it offers a one-sided comparison between two retrieval techniques. I say one-sided because the authors' analyses are so heavily focused on those cases where both algorithms are successful. However, they have clearly demonstrated that, for given sets of inputs, both algorithms are subject to large failure rates. The underlying causes of these failures are tantalizingly interesting and, I suspect, likely to be highly informative in developing the next iteration of automated aerosol typing algorithms.

This version of the manuscript also contains a number of flaws that make it difficult to understand. For example, a) important terms are never defined (e.g., color ratio); b) terms that are defined are used inconsistently (e.g., 'Ångström exponent', which is initially defined in terms of backscatter coefficients, is used interchangeably to describe backscatter-related Ångström exponents and extinction-related Ångström exponents); c) the descriptions of the classification algorithms are far too brief (e.g., the description of the NATALI algorithm does not provide the level of detail required to evaluate the sensitivity study that the authors perform); and d) the derivation of key results is not well-explained (e.g., the calculation required to produce the numbers reported in figures 4 and 5 are not described).

Below please find an annotated version of the manuscript the contains several more detailed comments and questions.

[revised manuscript text omitted]

---

## Referee Comment (RC2) · Anonymous Referee #2 · 18 Feb 2019

The manuscript applies two different algorithims to investigate the capability to correctly classify an aerosol layer based on the intensive optical properties.

To my opinion: - A clear demonstration of the aerosol classification methodology should be done in section 3.1 and 3.2 with a sketch. The aerosol properties used in each of the algorithms should be clarified in the manuscript. - A table with the aerosol classes finally used would be helpful along with the aerosol mixtures for each aerosol class - More complex cases should be also presented in the manuscript. - Clean continental cases are not well justified in the manuscript

Detailed corrections/suggestions is within the upload reviewed file.

[Figure]

Please also note the supplement to this comment:
https://www.atmos-chem-phys-discuss.net/acp-2018-1062/acp-2018-1062-RC2-supplement.pdf
* * *
[Figure]

**Supplement:**

[revised manuscript text omitted]

---

## Author Comment (AC1) · 22 Apr 2019

*As indicated by the title, this paper uses EARLINET 3α + 2β lidar measurements to compare the performance of two different aerosol typing algorithms: the Papagiannopoulos (AKA Mahalanobis distance) algorithm and the NATALI algorithm from Nicolae et al.*

*For the Papagiannopoulos algorithm, aerosol classes are defined using an ensemble of high quality EARLINET measurements that has been manually partitioned by subject matter experts to ensure uniformity of the optical characteristics of all observations within each class. For the Nicolae/NATALI algorithm, aerosol classes are defined using "values obtained from a specially designed aerosol model with simulations from over 50,000 synthetic cases of [different types of] aerosols". Both classification schemes are trained using supervised learning techniques (although this point is not mentioned in the manuscript).*
*The lidar measurements used as algorithm inputs for the Papagiannopoulos algorithm are*
*a) backscatter-related Ångström Exponent at 355nm and 1064nm (AE 355/1064 ),*
*b) the lidar ratio at 532 nm (LR 532 ), and*
*c) the ratio of the lidar ratios (LR 532 /LR 355 )*

*The lidar measurements used as algorithm inputs by the Nicolae/NATALI algorithm are not specified in this manuscript.*

*The high resolution variant of NATALI identifies 14 distinct aerosol types. The "low resolution without depolarization" variant used in this study recognizes five aerosol types: dust, smoke, continental polluted, continental, and maritime (page 6, line 15–16). For this study, the smoke and continental polluted types were combined into a single 'polluted smoke' type, so that only four types are recognized: dust, polluted smoke, continental, and maritime.*

*The standard Papagiannopoulos algorithm identifies 8 aerosol types: clean continental, polluted continental, pure dust, mixed dust (dust + maritime), polluted dust (dust + smoke and/or pollution), mixed maritime, smoke, and volcanic. For this study, these 8 types are combined to yield the same four types (page 7, line 1–2) that are recognized by the modified NATALI algorithm: dust (dust + volcanic + mixed dust + polluted dust), polluted smoke (smoke + polluted continental), clean continental, and maritime (mixed maritime).*

*Both algorithms were applied to the identical set of lidar measurements of 116 different aerosol layers. The NATALI algorithm successfully classified 80 of the 116 layers (~69%). The Papagiannopoulos algorithm successfully classified 114 of the 116 layers (~98%). The subsequent analyses are largely devoted to comparing classification results for the 80 cases in which both algorithms were successful.*

*The poor performance of the NATALI algorithm when applied to real-world lidar measurements suggests a disparity between the NATALI aerosol models and the EARLINET measurements.*
*Similar poor performance (39% failure rate) was found in a simulation study in which the authors used the NATALI aerosol model to generate inputs for the Papagiannopoulos algorithm.*
*Unfortunately, the authors do not investigate the possible causes or sources of these disparities.*

*In its present form, I cannot recommend this manuscript for publication. Most importantly, this study does not appear to address any specific science questions. Instead, it offers a one-sided comparison between two*

*retrieval techniques. I say one-sided because the authors' analyses are so heavily focused on those cases where both algorithms are successful. However, they have clearly demonstrated that, for given sets of inputs, both algorithms are subject to large failure rates.he underlying causes of these failures are tantalizingly interesting and, I suspect, likely to be highly informative in developing the next iteration of automated aerosol typing algorithms.*

The reviewer is right. In the manuscript we did not explained satisfactory the main objective of the paper and we did not investigate the possible causes of failure in typing. In the revised version of the paper these comments have been extensively taken into account discussing more these aspects (motivation, agreement and disagreement of the methods). Some text has been added about this and it is reported below into the replies to specific comments. Before addressing the specific comments we would like to give some general remarks about the revised version of the manuscript, which also aim to outline the science questions that are addressed in the paper.

Aerosol typing is an important topic recognized at international level as such: the AEROSAT (International Satellite Aerosol Science Network) initiative established a working group about aerosol types. This is because inferring the aerosol source is an important tool for identification of events, alerting of dangerous situations, definition of mitigation strategies. All these applications are of interest of wide communities. Many remote sensing, near surface and modeling communities have defined ways for inferring/estimating aerosol types and sources and there is the need for defining a common background on this (even nomenclature is not harmonized nowadays) and to compare aerosol type (which is not a numerical value) derived by different methods for improving common understanding and providing a sort of uncertainties on this quantity. In that context in our paper we try to address the following science questions:

- Is there a common understanding for the aerosol types between different typing schemes?
- Can automated methods provide operationally accurate typing when applied in data stored in a climatological database?
- How can we compare different typing methodologies? Which parameters affect the agreement or disagreement between them and the ability to provide typing results.

To address these questions we compare and test in this paper, two independently developed methods based on different approaches. The methods are only briefly described in the manuscript because the respective algorithm papers have already been published in literature. Up to now, there is not a well-established typing scheme and there is actually not an agreed method for evaluating such procedures. The comparison with manual analysis reported in the corresponding algorithm papers and here, however show a promising potential. The idea here is that the convergence of two different methods on the same type is a signature of reliability. This is the reason why emphasis was given to the cases where the two methods agreed. On the other hand, the reviewer is absolutely right: the cases where there was no agreement are extremely important for understanding the limitations of the assumptions behind the two schemes and provide a useful data set to investigate the causes, which will help to potentially improve the typing methods.

*This version of the manuscript also contains a number of flaws that make it difficult to understand. For example, a) important terms are never defined (e.g., color ratio);*

The reviewer is right. All the intensive optical parameters used in the comparison, are defined properly and all the equations are provided (Section 3) in the revised version of the text.

*b) terms that are defined are used inconsistently (e.g., 'Ångström exponent', which is initially defined in terms of backscatter coefficients, is used interchangeably to describe backscatter-related Ångström exponents and extinction-related Ångström exponents);*

The text is modified accordingly: "The intensive properties relevant to this study are: the extinction-related Ångström exponent (AE), the backscatter-related Ångström exponent (BAE), the ratio of the backscatter coefficients profiles (color ratios), the lidar ratio (LR), and the ratio of the lidar ratios (RLR). The respective formulas are provided in the following equations, where λ is the wavelength, z is the height, a is the aerosol extinction coefficient, and b is the aerosol backscatter coefficient."

*c) the descriptions of the classification algorithms are far too brief (e.g., the description of the NATALI algorithm does not provide the level of detail required to evaluate the sensitivity study that the authors perform); and*

A new paragraph with the description of the steps of the Mahalanobis distance-based typing algorithm is added, thus, providing detailed information about the reference dataset and the training of the classifier. Check our answer in Comment 26. Also, a paragraph describing the sensitivity test in NATALI software is added (see Comment 31).

*d) the derivation of key results is not well-explained (e.g., the calculation required to produce the numbers reported in figures 4 and 5 are not described).*

The revised version is changed accordingly, explaining the derived percentages. In Figure 5 percentages are replaced with numbers. All the numeric results, reported in figures 4 and 5 are now explained in detail in the text. See answer to comment 52 and 54.

*Below please find an annotated version of the manuscript the contains several more detailed comments and questions.*

We would like to thank the reviewer for his/her fruitful comments that led to the improvement of the manuscript. In the following, answers to comments are reported just below each related comment. When needed, the part of the manuscript we modified or added to the old version, is reported. Additionally, after taking into account the feedback from all the reviewers we proceeded to the following major changes in the revised version of the manuscript.

*1) Line 7 page 1 how do you know? what independent measurements or external metrics were used to determine typing success? I hope this topic is explained in detail in the main body of the manuscript. (note: algorithm convergence is not synonymous with successful typing.)*

The reviewer is right. We modified the sentence, which now reads: "71 layers out of 110 (percentage of 65%) were typed by both techniques and 56 of the 71 layers (percentage of 79%) were attributed to the same aerosol classes."

*2) Line 9 page 1 another note on "successful typing": the fact that both algorithms identify the same type does not mean that both algorithms are correct.*

The reviewer is right. We modified the sentence as follows: "The analysis showed that the two algorithms, when applied to real atmospheric conditions, provide typing results that are in agreement (88% for PollutedSmoke, 78% for Dust and 65% for CleanContinental)."

*3) Line 10 page 1 how different are the definitions of the types specified by the two algorithms? what degree of difference or similarity should we expect in the typing results based solely on the specifications for the different aerosol classes?*

In order to address this comment we added the following paragraph in section 4.4 of the revised version of the manuscript and a summary table (Table 3) which provides mean aerosol optical properties of the reference aerosol types used on the two automated algorithms.

Overall, there were 15 cases (on Thessaloniki dataset) that the two methods provided different typing results. There are seven cases typed as CleanContinental aerosol by NATALI and PollutedSmoke or CleanContinental by EMD, five cases typed as PollutedSmoke by NATALI and Dust by EMD, 2 cases typed as Dust by NATALI and CleanContinental by EMD and one case type as Maritime by NATALI and CleanContinental by EMD. These mismatches are illustrated in Figure 7. In order to understand these differences we highlight some critical issues relevant the type definitions of the two methods, based on Nicolae et al., (2018) and Papagiannopoulos et al., (2018).

**CleanContinental:** The contribution of the Soot component in the chemical composition of the CleanContinental category, allows higher lidar ratio values at 532nm (52–53sr) in the NATALI scheme. Consequently, layers recognized by NATALI as CleanContinental ones, in the EARLINET Mahalanobis distance-based typing method are attributed as signature for PollutedSmoke or Dust (as it can be seen in the revised Figure 6 -now Figure 7).

**Marine:** As observations of pure maritime particles are quite scarce within EARLINET and, generally, when these particles are observed, their characteristics are far from pristine, the Maritime category for the EARLINET Mahalanobis distance typing algorithm**,** corresponds to mixed maritime layers. This is different from the pure maritime category that NATALI identifies. Lower lidar ratio values for 532nm (19–25sr) are defined in the NATALI software for the identification for marine layers, in contrast to the higher ones (16-32sr) allowed in the EARLINET Mahalanobis distance typing algorithm.

**Dust:** Higher values of Lidar ratio at 532nm are allowed for EARLINET Mahalanobis distance typing algorithm in identifying dust particles (48-62sr), considering all dust-like aerosol types as one category, while NATALI allows values corresponding to more pure cases (44–49). Therefore, a number of Dust recognition are attributed to PollutedSmoke particles (6 cases) or CleanContinental particles (1 case) in the NATALI output.

**PollutedSmoke**: An almost perfect score was found for the PollutedSmoke category. This can be attributed to the similar reference values attributed by both typing algorithms. Higher values of Lidar

ratio at 532nm are allowed for NATALI (62-92sr - ContinentalPolluted and Smoke), instead of the lower ones (52–89sr) allowed in the EARLINET Mahalanobis distance typing algorithm.

*4) Line 16 Page 1 why? what information does aerosol classification provide? is this information critically important all by itself, or solely as input to other applications (or both)? please explain. while AMT is a technology journal, ACP is a science journal. it is therefore essential to clearly explain the scientific importance of aerosol classification.*

The reviewer is right. The scientific importance of aerosol classification schemes should be provided in the text. Therefore, we modified the first paragraph of the Introduction, which now reads:

"Aerosol classification is a key parameter in understanding the impact of different aerosol sources on climate, weather systems, and air quality. Given that, aerosols are emitted from various sources, a significant number of aerosol types with different optical and microphysical properties coexist in the atmosphere. As a consequence, considerable uncertainties have been raised in the quantification of aerosols in the radiative transfer calculations and their interactions with clouds, leading the interest of wide communities for improving common understanding related to aerosol types and sources.

Nowadays, aerosol typing is an interesting topic recognized at international level, as inferring the aerosol source is an important tool for identification of events, alerting of dangerous situations, definition of mitigation strategies. The International Satellite Aerosol Science Network (AEROSAT), for example, is an initiative established from different research groups around the world focusing on aerosol retrieval. AEROSAT identifies the need of a common background for aerosol typing and typing procedure's comparison efforts as one of the main important task to be tackled into the aerosol community (Mona et al., 2015)."

Also, the following sentences are added in the introduction:

"Therefore, the need for defining a common background on aerosol types (even nomenclature is not harmonized nowadays) and for comparing aerosol types (which is not a numerical value) derived by different methods is on the top of interest."

"This paper can be considered as a step towards the general objective of finding translating rules and way for quantifying differences between typing procedures."

Also, the following sentences are added in the Sect. 5

"Considering that up to now, there is not a well established method to provide correct results under every conditions, the systematic and full comparison of different methods is the only way to have a better insight about the possibility of inferring the aerosol source from the observed optical properties. Additionally, the modularity of the EARLINET Single Calculus Chain tool offers the possibility to implement aerosol typing procedures for automatically providing the aerosol typing information. A comparison between the two existing typing schemes that make use of the aerosol optical properties could be rather useful. This will allow also from one side the optimization of aerosol

property models and from the other side the definition of the aerosol types used and the selection of the reference dataset."

*5) Line 18 Pag. 1 do you really mean "magnitudes" (which can be directly measured) rather than "mechanism" (which must be inferred)*

The text has been modified to:
"Thus, a number of typing schemes have been applied to classify aerosols, based on the synergistic information of the backscattering and extinction coefficients and the polarization state of the received light at different wavelengths (e.g, Ansmann and Müller, 2005, Groß 2013)."

*6) Line 19 Page. 1 according to Hamill et al. (2016), "Our aerosol characterization is based on the AERONET retrieved quantities, therefore it does not include low optical depth values." somewhere in this manuscript, I hope the authors will discuss the range of optical depths over which each method can reliably ascertain aerosol types.*

The reviewer is right. When examining the ability of the methods to provide a classification result for a certain aerosol type or mixture the uncertainties of the measurements have to be taken into account. As large measurement uncertainties prevent a correct aerosol type separation, high quality measurements are mandatory. Some criteria apply for the layers identification (Belegante et al., 2014). In particular, the relative error of each intensive optical parameter calculated as mean within the layer should be lower than 50 % in order to be considered for the typing procedures. Additionally, the range of the parameters to ensure reliable retrievals of aerosol types, are discussed in terms of the values of the extensive or intensive parameters that were used in the typing modules. So, a table with the acceptable limits for the average intensive optical parameters was added in the text. In addition, we examined the ability of the both methods to provide typing results, relative to the derived layer AOD values. This study showed that the mean layer AOD of the typed layers has similar value for both typing techniques (0.1076 ± 0.0898 (SD) for Natali and 0.11166 ± 0.0928 (SD) for the EARLINET Mahalanobis distance typing algorithm (EMD). However, the EARLINET Mahalanobis distance algorithm seems more sensitive to lower values (not typed cases show a mean layer AOD of 0.0298 ± 0.0578 for EMD, while 0.06899 ± 0.0606 for NATALI). This information has been included in the revised mansucript.

*7) Line 5 Page. 2 In addition to these examples, please provide definitions for both intensive and extensive the key difference is that extensive properties vary with aerosol concentration (or amount or loading), whereas intensive properties are independent of concentration and instead vary only according to aerosol composition (i.e., type). it should also be noted even intensive properties might not be sufficient to guarantee accurate typing, as some aerosol types (e.g., biomass burning and industrial pollution) have very similar intensive properties but very different sources and/or generating mechanisms. these definitions should be provided the first time the terms are used.*

We thank the reviewer for this comment. The text included is given below:

"The extensive properties depend on the aerosol concentration, whilst intensive ones are type sensitive and provide separate classification for each detected layer (e.g., Müller et al., 2007; Ansmann et al., 2011; Tesche et al., 2011; Burton et al., 2012; Pappalardo et al., 2013; Groß et al., 2013; Amiridis et al., 2015; Giannakaki et al., 2015; and Baars et al., 2016). Nevertheless, even intensive properties might not be sufficient to guarantee accurate typing, as some aerosol types (e.g., biomass burning and industrial pollution) have very similar intensive properties but are attributed to different sources and generating mechanisms."

*8) Line 10-11 Pag. 2 also CATS; see Yorks et al., 2016: CATS Algorithm Theoretical Basis Document, Level 1 and Level 2 Data Products, https://cats.gsfc.nasa.gov/media/docs/CATS_ATBD.pdf*

*references? (e.g., Burton et al., 2012)*

*references? (e.g., Papagiannopoulos et al., 2018)*

Thank you for this comment. The appropriate references have been added.

"Aerosol typing schemes have been developed for high-resolution lidar measurements of space-borne lidars (e.g., CALIPSO, Cloud-Aerosol Lidar and Infrared Pathfinder Satellite Observation, Omar et al. 2009; EarthCARE, Illingworth et al. 2015, CATS, Yorks et al., 2016), airborne - High Spectral Resolution Lidar -HSRL- measurements (e.g., Burton et al., 2012; Groß et al., 2013) and multiwavelength Raman measurements (e.g., Nicolae et al., 2018; Papagiannopoulos et al., 2018)."

*9) Line 26 page 2 I'm impressed by this very extensive and complete list of references*

We thank the reviewer for acknowledging the efforts of EARLINET.

*10) Line 4 Page 3 this statement needs references. off the top of my head, I would expect the extinction Angstrom exponents of volcanic ash and dust might be quite different.*

Yes, it is true that volcanic ash and desert dust can have different sizes depending on the transport pattern and aging processes. Indeed, many differences can be observed in the optical properties even among volcanic particles depending on the specific volcano and the eruption type and among desert dust particles  depending on the source region. What we would like to underline here is that volcanic and desert dust particles are both characterized by the asphericity of the particles, observable and measured through the linear depolarization ratio (while typically the other aerosol types have small values of this parameter), the other intensive optical properties could be similar in ranges e.g Angstrom exponents and lidar ratio (see for example fig. 5 in Nicolae et al., 2018; fig. 7 in Papagiannopoulus et al., 2018 and the variety of volcanic values in Mona et al., 2012). However, because the source region for these cases is well defined the use of transport models could allow for a more precise attribution of the aerosol source. EARLINET-based studies for the Eyjafjallajökull eruption have indicated similar Ångström exponent values to desert dust. Specifically, Papayannis et al., (2012) found mean Angstrom exponent values of 0.57 ± 0.26. Furthermore, Sicard et al. (2012) and Ansmann

et al. (2010) calculated a mean Ångström exponent of 0.68 ± 0.63 and 0.03±0.4 for volcanic ash respectively. Following the suggestion of the reviewer, references have been added:

"Furthermore, the combined use of lidar observations and transport model simulations permit the discrimination of desert dust and volcanic ash particles that typically, have the same optical characteristics (see for example fig. 5 in Nicolae et al., 2018; fig. 7 in Papagiannopoulus et al., 2018 and the variety of volcanic values in Mona et al., 2012). Simultaneous observations of desert dust and ash particles were made during the Eyjafjallajökull volcanic eruption in 2010 and the methodology for the type discrimination was presented by Papayannis et al. (2012); Mona et al. (2012); and Pappalardo et al. (2013)."

*11) Line 8 Page 3 what science questions are being addressed by this comparison?*

*As stated also in our general remarks the aim of this comparison is to check the performance of the two techniques to a specific environment, focusing on the reasons that they agree or disagree and the limitation of each in the aerosol typing. Check the performance of each different reference dataset is really important for defining a common understanding on aerosol types (even nomenclature is not harmonized nowadays) and for evaluation strategies to compare aerosol types (which is not a numerical value) derived by different methods. In that context the applied thresholds, differently introduced in each algorithm, are a critical factor in the identification rate of the two algorithms. The latter is discussed in more detail in the revised manuscript.*

The next paragraph is added to the manuscript and we think that pinpoints the importance of this work:

"Aerosol typing schemes have been discussed to be implemented in the next version of the EARLINET Single Calculus Chain tool (SCC; D'Amico et al., 2015). SCC was developed for the analysis of the data of different lidar systems in an automated, unsupervised way and currently SCCv5.0.11 delivers profiles of optical aerosol properties. Future work is dedicated to the implementation of new features like profiles of intensive optical properties, detection of aerosol layer geometrical properties and calculation of the intensive optical properties per detected layer, which will further allow the classification of the observed layers into aerosol types. Therefore, the need for defining a common background on aerosol types (even nomenclature is not harmonized nowadays) and for comparing aerosol types (which is not a numerical value) derived by different methods is on the top of interest.

Given the aforementioned status, a first attempt of comparing and evaluating two classification tools developed within EARLINET, that provide near-real-time aerosol typing information for the lidar profiles of Thessaloniki, is presented. Our aim is: (i) to check the performance of both supervised learning techniques in their low resolution mode, which is the case for the majority of the available measurements within EARLINET, (ii) to highlight the strong and weak aspects of both algorithms when applied to lidar data from a station where, typically, variable mixtures of aerosols are present and (iii) to investigate the reasons of typing agreement and disagreement with respect to the uncertainties and the threshold criteria applied. This paper can be considered as a step towards the general objective of finding translating rules and way for quantifying differences between typing procedures.".

*12) Line 23 Page 3 unclear; were particulate depolarization ratios previously available? or have the detectors and optics only recently been added, and hence the depolarization measurements are not yet part of the standard data stream?*

The two depolarization channels have been added to measure the cross and parallel polarized signal at 532 nm, but due to issues, related to the polarization purity of emitted laser beam and the performance of the telescope lenses, the particle depolarization ratio was not available. A new telescope was installed at the end of 2017, thus for the period under study no depolarization measurements are available. Indeed, these issues could also partially affect the total 532 nm elastic backscatter measurements

So, according to the above, the text has been modified to:
"These two channels have been added to measure the cross and parallel polarized signal at 532 nm, but due to technical issues the particle depolarization ratio is available after August 2018."

*13) Line 2 Page 4 why is depolarization enclosed in parentheses? if depolarization ratios are optional but not actually required, they shouldn't be cited as requirements.*

The reviewer is right. The text has been modified to:
"The two automatic aerosol typing methods require only lidar data with 3β + 2α (3 backscatter and 2 extinction coefficient profiles) configuration without any use of ancillary external information."

*14) Line 2 Page 4 this seems like an odd choice to me. why not use the extinction Ångström exponents instead? surely the extant literature contains much, much more information about extinction Ångström exponents (which can be obtained by any number of sensors and are of interest in a wide variety of research applications) than about the ratio of lidar ratios (which are only of direct interest to lidar practitioners).*

The ratio of lidar ratios (RLR) is a quantity that provides information about the aerosol size, and is preferred due to the fact that, the extinction coefficient, is associated with larger uncertainty. This is attributed mainly to the derivative in the inversion algorithm (e.g, Ansmann et al., 2002; Freudenthaler et al., 2009). Also, the Ratio of Lidar Ratios was chosen as a classification parameter, after the performed sensitivity analysis from Papagiannopoulos et al., (2018), to identify which classifying properties provide the adequate information to better predict the correct aerosol class. Two statistical parameters were used in the analysis: the total and the partial Wilks' lambda (3; Wilks, 1963) that are widely used, e.g, Burton et al. (2012) and Russell et al. (2014).
RLR quantity is not only available by lidar systems, but can also be obtained by the Sunphotometers of the Aerosol Robotic Network (AERONET), by dividing the provided lidar ratios, (Shin et. al., 2018).

In the revised paper the introduction of Section 3 reports all the definition and meaning of the different parameters and then it is shortly recalled the work done in Papagiannopoulos et al., 2018 for selection of how many and which parameters are the most suitable from a mathematical point of view for the typing purposes.

"The intensive properties relevant to this study are: the extinction-related Ångström exponent (AE), the backscatter-related Ångström exponent (BAE), the ratio of the backscatter coefficients profiles (color ratios - CI), the lidar ratio (LR), and the ratio of the lidar ratios (RLR). The respective formulas are provided in the following equations, where λ is the wavelength, z is the height, a is the aerosol extinction coefficient, and b is the aerosol backscatter coefficient."

*15) Line 10 page 4 please provide the equation*

All the equations used for the typing are included in the text (in Sect. 3 "Aerosol Typing Methods").

*16) Line 10 page 4 wouldn't extinction Ångström exponents be a more accurate and reliable size indicator? (certainly extinction Ångström exponents have been investigated much more thoroughly in previous publications than have backscatter Ångström exponents).*

As extinction coefficient, is associated with larger uncertainty, the backscatter-related Ångström exponent is preferred. There is a number of studies that have used in their aerosol classification this parameter, e.g., Müller et al. (2007).  Besides, the backscatter related Ångström exponent  has been extensively used  in the climatological analysis made for Thessaloniki, EARLINET station, by Amiridis et al., (2009), Gianakkaki et al. (2010), and Siomos et. (2018).

*16) Line 20 page 4 what are color ratios? please define this term.*

The term is now described in the revised manuscript. The text has been modified to:
"The extinction-related Ångström Exponent at 550nm and 350nm (AE 350/550), the lidar ratio at 350 nm (LR 350), the lidar ratio at 550 (LR 550), and the ratio of the backscatter coefficients profiles (color ratios) CR 350/550 and CR 550/1000 , are used for the aerosol classification."

*18) Line 23 page 4  this seems misleading to me. according to Nicolae et al. (2018), the required data are "backscatter coefficient (β) profiles at 1064, 532 and 355 nm, extinction coefficient (α) profiles at 532 and 355 nm, and, optionally, linear particle depolarization (δ) profile at 532 nm."*

The reviewer is right and indeed this was not clear in the manuscript. The next few lines are included in the revised manuscript:
"The input module requires optical properties profiles as those measured by EARLINET stations, namely the aerosol extinction coefficient and the aerosol backscatter coefficient profile. Optionally, the linear particle depolarization profile at 532 nm can be provided, so as to allow a better classification and to increase the number of classified aerosols types."

Also, the following paragraph is modified:

Depending on the availability of the particle linear depolarization ratio and the quality of the provided lidar profiles, the derived typing can be either of high resolution (AH), or low resolution with depolarization (AL) or low resolution without depolarization (BL). Pure aerosols categories, and even mixtures of three aerosols types can be obtained from the NATALI algorithm. In the high resolution typing, 14 aerosol types can be distinguished (i.e., Continental, ContinentalPolluted, Dust,

Maritime/CC, Smoke, Volcanic, Coastal, CoastalPolluted, ContinentalDust, ContinentalSmoke, DustPolluted, MaritimeMineral, MixedDust and MixedSmoke) when the quality of the provided optical products is high enough. In the low resolution typing (AL), 6 predominant aerosol types can be provided but with high uncertainty (i.e, Continental, ContinentalPolluted, Smoke, Dust, Maritime and Volcanic). The low resolution typing (BL) provides 5 predominant aerosol types (Dust, ContinentalPolluted, Smoke, Continental, Maritime) either pure or mixed, when the depolarization information is not provided. Finally, the output module provides the intensive optical parameters within each layer along with their mean value and the corresponding uncertainty.

*19) Line 26 page 4 I'm not sure I understand what's meant here. are the authors computing weighted means, where the weights are the SNRs at each data point? if so, won't in-layer attenuations and the attenuation of the signals between successive layers adversely affect the SNR of the points at higher altitudes, thereby biasing their contributions to the mean values?*

No. We mean that we calculate mean properties only for layers where the SNR is greater than 5. The text is changed accordingly.

*20) Line 30 page 4  either "most likely" or "most probable", but not "most likely probable"*

This was a typo and it is now fixed.

"The identification of the most probable aerosol type is then made through a voting procedure using the results from the three ANNs interrogated."

*21) Line 32 page 4 since this is a "specially designed aerosol model", why does it use wavelengths that are different from the measured wavelengths?*

The text has been modified accordingly:
"The calculations of the optical properties for each layer are used in the typing module and compared to values obtained from a specially designed aerosol model with simulations from over 50000 synthetic cases of aerosols. Specifically, the synthetic database was developed using the aerosol model builted for 350, 550, and 1000 nm sounding wavelengths, based on the 61 wavelengths of OPAC (Optical Properties of Aerosols and Clouds) software package (Hess et al., 1998), for which the microphysical characteristics of the aerosols are available from GADS. These wavelengths are then re-scaled to the usual lidar wavelengths (i.e., 355, 532 and 1064 nm) using an average Ångström exponent equal to one. This was considered a valid assumption for all aerosol types, taking into account the small difference between the lidar and the model wavelengths. A comprehensive description of the developed aerosol model can be found in Nicolae et al., (2018)."

*22) Line 32 page 4 need to define color ratio*

The term is now described in the revised manuscript (also the equation is provided in Section 3).

"The extinction-related Ångström Exponent at 550nm and 350nm (AE 550/350 ), the lidar ratio at 350 nm (LR 350 ), the lidar ratio at 550 (LR 550 ), and the ratio of the backscatter coefficients profiles (color ratios) CR 550/350 and CR 1000/550 , are used for the aerosol classification."

*23) Line 32 page 4 presumably the 350 nm, 550 nm and 1000 nm wavelengths are specified by the model. please state definitively which lidar measured parameters are used in the configuration of the NATALI algorithm used in this paper.*

The next few lines are included in the revised manuscript:

"The synthetic database was developed using the aerosol model built for 350, 550, and 1000 nm sounding wavelengths, based on the 61 wavelengths of OPAC (Optical Properties of Aerosols and Clouds) software package (Hess et al., 1998), for which the microphysical characteristics of the aerosols are available from GADS (Global Aerosol Date Set). These wavelengths are then re-scaled to the usual lidar wavelengths (i.e., 355, 532 and 1064 nm), that are used as input to NATALI, using an average Ångström exponent equal to one."

*24) Line 9 page 5  so all NATALI results used in this comparison study will be BL only, yes?*

Yes, the comparison is made for the low resolution without depolarization (BL). A Table is added to provide the information of the resolution and the aerosol types that constitute the classes investigated in this study.

*25) Line 9 page 5  do you mean "along with" here?*

The reviewer is right. The sentence now becomes:

"Finally, the output module provides the intensive optical parameters within each layer along with their mean value and the corresponding uncertainty."

*26) Line 25 page 5 seems like you're omitting an important step here. see section 3.2 in Papagiannopoulos et al., 2018. before you can start classifying previously unclassified data, don't you first have to train your classifier to recognize the manually specified classes in the reference dataset established in step 1?*

The reviewer is right. The text has been modified to:
"The classification of each aerosol layer is made by calculating the distance of an observation from the defined reference classes and by attributing each observation to a specific class based on the minimum distance. Given that the overall predictive accuracy of a typing algorithm, depends on the defined reference classes, the choice of the appropriate reference dataset consists a crucial parameter. So, a well-characterized EARLINET dataset of observations from 2008 to 2010 was used to define the aerosol classes (Pappalardo et al. 2013; Papagiannopoulos et al. 2016a, and Schwarz 2016). As this dataset did not include all the typical aerosol components presented over Europe, the observations were enhanced with additional ones (already published in literature), contributing to a total of 69 layers (Pappalardo et al., 2013; Papagiannopoulos et al., 2016a), corresponding to the

following types: CleanContinental (9 cases), PollutedContinental (16 cases), Marine (8 cases), Dust (9 cases), MixedDust (10 cases), PollutedDust (5 cases), Smoke (7 cases), Volcanic ash (5 cases).

In a next step, a sensitivity analysis was performed in order to identify the classifying properties that provide the most adequate information to the better prediction of the correct aerosol class. The intensive properties that held the most weight among others in the classification appeared to be the backscatter-related Ångström Exponent at 355nm and 1064nm (BAE 355/1064 ), the lidar ratio at 532 nm (LR 532 ), and the ratio of the lidar ratios (LR 532 /LR 355 ). Then, the initial reference dataset was split into a training and a validation dataset, in order to evaluate the predictive accuracy of the automatic method. Therefore, the training dataset was used to calculate the classification functions and then each observation was validated using the training dataset. Additionally, the effect of the depolarization information to the training phase was also investigated. To complement the reference dataset in this context, general literature values for particle linear depolarization ratio at 532 nm were used. Generally, the depolarization ratio is a really important parameter that can strengthen the typing procedure in the presence of aspherical layers, but Papagiannopoulos et al. (2018) showed that for the 5 and 4 classes, the particle linear depolarization ratio became less important (in this case the highest weight in the classification corresponds to the lidar ratio at 532 nm)."

*27) Line 25-30 page 5 the description here does not provide enough detail for me to grasp the essentials of the algorithm. please expand this section.*

See above, the answer to Line 25 page 5.

*28) Line 31 page 5 what is the physical meaning of this "normalized probability" and how does one compute it? this term should be clearly defined in the manuscript.*

*also, I'm assuming that the authors mean that "normalized probabilities" are computed for all candidates, and that for a successful classification to occur, the largest "normalized probability" must exceed 0.5. please add text to clarify.*

The Mahalanobis distance quantifies the distance between a point (i.e., the unclassified measurement) and a distribution (i.e., the aerosol class). This distance is estimated for each of the available number of classes and, thus, it is independent. Besides, the Mahalanobis distance gives an estimate of the probability for each aerosol class. This probability has to be normalized so as to provide an estimate of a relative probability for each class. The normalized probability ensures that similar class distances will not be classified. The following equation is used to derive the normalized probability:

$$\frac{\frac{1}{d^2}}{\sum \left(\frac{1}{d^2}\right)}$$ , where d stands for the distance. More details can be found at Burton et al. (2012). A

proper explanation is included in the revised manuscript.

29) Line 6 page 6 classifications showed a high success rate

The text has been modified to:
"Dust classifications showed a high success rate, whilst the aerosol types that performed worse were the smoke and polluted continental aerosol."

30) Line 19 page 6 should this be 'minimum layer thickness'?

The reviewer is right. This should be the minimum layer thickness.

31) Line 19 page 6 the authors need to provide the following information before readers (and reviewers) can evaluate the effectiveness and completeness of their sensitivity analysis.

(a) where and how is the layer thickness parameter used?
(b) what is being smoothed and where (and how often) in the analysis it this smoothing applied?
(c) what quantities are the minimum confidence and minimum agreement ratio applied to?

(a) Layer thickness parameter is defined by the user in the graphical interface of NATALI module. The parameter is defined in meters, so as NATALI module to reject layer sub-structures thinner than the selected value.

(b) Smoothing is applied to the derivative of the 1064nm signal for the definition of the boundary layers. The inflexion points of the second derivative of the profile data, computed with the Savitzky-Golay filter, give the boundaries of the layers. The window size of the cubic Savitzky-Golay filter, which can be modified by the user, has a default value of 700 m. The filter was applied twice to obtain the second derivative. A signal- to-noise ratio filter is applied at this point, making sure the said ratio is at least 5. The layer boundaries are moved towards the median height until the SNR criteria is met; if the criteria cannot be satisfied with a layer height greater than e.g., 300m, the layer is discarded. Gross or fine structure of the aerosol layers is revealed by a higher or lower value of the adjustable smoothing parameter (FINESSE).

(c) The identification of the most probable aerosol type is made through a voting procedure, using the results from the three ANNs interrogated. Over 50000 aerosol synthetic data have been used to train the ANN and identify the better ANNs to classify the aerosols type from multiwavelength lidar data. The capability of ANNs to resolve the overlapping clusters of the intensive optical parameters is used on NATALI algorithm. The answer selected is based on a statistical approach. The selected types of ANNs classify the aerosols based on the response with high i) confidence (i.e. the probability of having one of the aerosol types) and ii) stability over the uncertainty range (i.e., the percentage of agreement for values between error limits). Therefore, answers with low confidence are filtered out and NATALI returns the 'Unknown' type. In this study, we select initially the confidence level for the output retrievals higher than 0.9 (minimum accepted confidence) and the minimum agreement threshold as default (i.e., 0.25).

The text has been modified accordingly in the revised manuscript.

*32) Line 20 page 6 I'm afraid I have no idea what this sentence is supposed to tell me. this is the first mention of smoothing in the manuscript. what is being smoothed, and why are you smoothing it? 700 number of values generated between value ± uncertainty" sounds like the description of a simulation study. how does this relate to smoothing?*

The reviewer is right. This is stated wrongly and not related with the smoothing. The selected (from the user) number of values (10-50) generated between [mean intensive optical parameter – uncertainty] and [mean intensive optical parameter + uncertainty]  correspond to finesse parameter.
Optical parameters calculated from lidar measurements are reported in the EARLINET database with the mean value  and the associated uncertainty, but the optical parameters calculated from synthetic data do not "carry" this uncertainty. For this reason, a fixed relative error was considered, which was multiplied with the mean value to obtain the absolute error (uncertainty). For the actual retrieval of the aerosol type, any value between within mean value +/-uncertainty was possible; therefore the algorithm was applied for all these values with a certain step (i.e. the finesse). The output is a "bundle" of possible aerosol types, with a dimension equal to the finesse.
The text in the revised manuscript is changed accordingly.

*33) Line 20 page 6  I'm bewildered by the repeated use of the word "value". do you mean the number of in-layer values lying between the characteristic mean ± uncertainty for each parameter? how does this relate to smoothing?*

Indeed, values correspond to the number of in-layer values lying between the characteristic mean parameter ± uncertainty.

*34) Line 33-34 page 6 these two statements contradict one another. please clarify: does the NATALI algorithm identify 5 types or 4 types?*

The reviewer is right. It was not stated clearly in the text. NATALI algorithm output in the low resolution module identifies 5 types. We decided to merge the two categories -PollutedContinental and Smoke-  to a more generic one, the category PollutedSmoke. The selection of four aerosol classes stems from the difficulty to distinguish types of aerosols that reflect the same aerosol characteristics. Considering the aforementioned typing merging, the EARLINET Mahalanobis distance-based typing algorithm was set to classify observations into 4 aerosol classes. This information is summarized and made more clear in Table 2 of the revised manuscript.

So, the text has been modified accordingly:

"In what follows, we merged the output types from NATALI that tend to reflect the same aerosol characteristics, and hence we evaluate the corresponding effects on the prediction rate of the algorithms. Thus, the smoke and the polluted continental categories were grouped into the more generic type of small particles with high lidar ratio values. The selection of four main aerosol classes stems from the availability of intensive properties, the difficulty in deriving a confident classification

without particle linear depolarization ratio and the difficulty in discriminating polluted continental and smoke particles that reveal the same type characteristics. Regardless, the aerosol classes describe the major aerosol components."

And later on the methodology part, the following change is made:

"The identified layer boundaries from NATALI are used as input in the EARLINET Mahalanobis distance-based typing algorithm. Considering the aforementioned typing merging, the EARLINET Mahalanobis distance-based typing algorithm was set to classify observations into 4 aerosol classes: CleanContinental, Dust, Maritime and PollutedSmoke."

*35) Line 1 page 7 please provide your rationale for combining types as you've done. Line 1 page 5 again, please provide your rationale for combining these two types. also, to avoid confusing your readers, this information should be given earlier (e.g., around line 15 on the previous page)*

We merged types that tend to reflect the same aerosol characteristics; the smoke and the polluted continental category were grouped into the more generic type of small particles with high lidar ratio values. Their intensive properties tend to be overlapped as can be seen in Groß et al. (2013). See also, answer 34. Also, in the low resolution typing all the dust like particles are considered as one category, given that the depolarization ration is not available to distinguish between pure dust and mixed/polluted dust particles. Also, a Table is added to provide all the necessary information.

*36) please provide a reference. I know of references supporting the assertion of low extinction Ångström exponents for dust (i.e., close to zero, as in Freudenthaler et al., 2008: doi:10.1111/j.1600-0889.2008.00396.x), but I can't immediately recall any for backscatter Ångström exponents.*

Müller et al. (2007) provided an analysis for backscatter Ångström exponents, reported values of 0.2 ± 0.2 for desert dust events. The following sentence was added:

"Also, Müller et al., (2007) reported values of mean backscatter-related Ångström exponent BAE 355/1064 of 0.2±0.2 for desert dust events."

*37) I note that table 4 in Siomos et al. 2018 reports backscatter Ångström exponents for dust mixes in the neighborhood of ~1. a "low Ångström exponent" would be ~0.*

The reviewer is right. These values reported by Siomos et al. (2018) correspond to mixed dust cases (and not pure one) typically observed over the urban Thessaloniki station.

*38) Line 13 page 7 see my previous comment. table 4 in Siomos et al. 2018 reports backscatter Ångström exponents for dust mixes of ~1. values of ~1 cannot simultaneously be characterized as both "low" and "relatively high"*

The reviewer is right and the text is modified to:

"Generally, continental particles (i.e., small with medium lidar ratio values) present low lidar ratio values, (i.e., 20–40 sr) and values of Ångström exponents around 1.0–2.5."

*39) Line 19 page 7 were the same criteria used in the automated typing also used in the manual typing? what (if any) additional information was used in the manual typing exercise that was not available in the automatic typing schemes?*

The so-called manual aerosol typing can be described by a three-step procedure (Papagiannopoulos et al., 2016). First, the aerosols and clouds are identified in the lidar data. Second, the boundaries of the aerosol layer are defined. Third, the aerosol layer is typed by means of investigation of intensive optical properties, model outputs (e.g., HYSPLIT simulations), and ancillary instrument (e.g., AERONET sunphotometer data, MODIS FIRMS data). Both automatic methods make use of lidar intensive properties and predict the aerosol type with no further external information. The lidar measurements of intensive properties are shown to vary with aerosol type and are able to provide an objective aerosol characterization.

*40) Line 4 page 8 the authors need to do a much better job of defining terms and using them consistently. what's the difference between AE355-532 and BAE355-532? what's the difference between AE355-532 and AE355/532? on page 4, line 10, AE is defined as the backscatter-related aerosol Ångström exponent. at this point in the manuscript, no definition of BAE has been given. scanning ahead to page 8, line 19, I see that BAE is also defined as the backscatter-related aerosol Ångström exponent.*

The reviewer is right. The definitions of the backscatter-related aerosol Ångström exponent and the extinction-related aerosol Ångström exponent were not provided clearly in the text. In the revised version, though, everything is stated properly and all equations are provided in Section 3.
The text has been modified to:
"The intensive properties relevant to this study are: the extinction-related Ångström exponent (AE), the backscatter-related Ångström exponent (BAE), the ratio of the backscatter coefficients profiles (color ratios), the lidar ratio (LR), and the ratio of the lidar ratios (RLR). The respective formulas are provided in the following equations, where λ is the wavelength, z is the height, a is the aerosol extinction coefficient, and b is the aerosol backscatter coefficient."

*41) Line 4 page 8 the Papagiannopoulos algorithm used the 355-to-1064 nm backscatter-related Ångström Exponent. why are those values not given here? why are the 532-to-1064 nm backscatter-related Ångström Exponent given? this is the first time they're mentioned in the manuscript, and it's not apparent that they're used in any aspect of either classification scheme.*

The reviewer is right. The 355-to-1064 nm backscatter-related Ångström Exponent is provided in the plots in the revised version. The 532-to-1064 nm backscatter-related Ångström Exponent was provided in the plots, because this was one of the optical parameters considered for typing in NATALI.

*42) Line 8 page 8 is this a "normalized" probability? The probabilities shown in the figure for the different types do not appear to add up to one. why not? please add text to explain.*

The probability reported here is not normalized, but the chi-square probability for each class. This probability is assigned to each distance, it is independent, and shouldn't add up to one. See previous comment (n 28).
The following paragraph is added:

"The output of the algorithm is the calculated Mahalanobis distance of the layer from each aerosol class and the calculated chi-square probability. For improving the reliability of the output, two screening criteria are applied to the minimum distance following the procedure of Burton et al. (2012). In our study, we use 3 classifying parameters (i.e., the type dependent intensive properties) and the minimum accepted distance for a measurement to be labelled is 4. When the distance is higher than the defined threshold, which means that no similarity with the aerosol classes of the reference datasets is found, the observation is not typed. Secondly, an extra screening criterion is applied to almost equal values for more than one aerosol class. The normalized probability (this probability is based on the calculated distances) of the aerosol class needs to be higher than 50%."

Also, the following sentence is added to Section 4.1 and 4.2.
"The plotted probabilities are assigned to each calculated distance and are different from the normalized ones that used as a screening criteria (and add up to 100)."
Similar comment is made in the captions of Figures 1, 2: 'EARLINET Mahalanobis distance-based typing algorithm probability for each detected layer (different from the normalized one)"

*43) Line 18 page 8 these look like very representative value for the extinction Ångström exponent of smoke (though layers B and C are slightly low). perhaps it's worth adding a reference that established a connection to previously reported values?*

Tesche et al. (2011) reported values for the extinction Ångström exponent of smoke 0.9 ± 0.26 . The following sentence is added:
"These values are consistent with previous reported ones, e.g., Tesche et al. (2011a) reported values of the extinction Ångström exponent for smoke of 0.9 ± 0.26."

*44) Line 20 page 8 again, why report BAE532-1064 when the Papagiannopoulos algorithm uses BAE532-1064? as a referee (and as a reader!) I'm looking for correspondence between the measurements and the characteristic attributes of the aerosol types defined by each algorithm.*

The reviewer is right. The 355-to-1064 nm backscatter-related Ångström Exponent was provided in the plots in the revised version. The  355-to-1064 nm backscatter-related Ångström Exponent was provided, because this parameter is one of the parameters taken into account for NATALI typing.

45) Line 30 page 8 so, the EARLINET-Mahalanobis algorithm achieves an almost perfect score (114/116), whereas NATALI fails ~30% of the time? that alone is a very notable performance difference between the two methods. where is the analysis that explains this stark difference? in particular, how do the "manually typed" results compare to the EARLINET-Mahalanobis algorithm classifications for the 34 layers for which NATALI failed but the EARLINET-Mahalanobis algorithm succeeded?

Indeed, the two methods are different. This is general problem of the various aerosol typing algorithms and as already discussed in our general remarks, the international initiative AEROSAT identifies the need of a common background and comparison efforts as one of the main important task to be tackled into the aerosol community and a working group was set up for this purpose. One of the coauthor of this paper is leading this Working group and this paper can be considered as a step towards the general objective of that WG for finding translating rules and ways for quantifying differences between typing procedures. In particular, the Mahanalobis distance typing is designed in such a way that it does not take into account the error on the measurements. It considers the mean measured values as true and when the distance of the measurement from the clusters defined through the reference datasets is large, with respect to a defined threshold, the observation is not typed at all. Comparing methods which are too similar indeed would not add any information to the scene, what is instead interesting is understanding where the 2 methods converge and where they see differences. There is not nowadays one established method which provides accurate results under every condition, so the systematic comparison of different methods is the only way to have a better insight about the possibility of inferring the aerosol source from the observed optical properties.

However it should be pointed out, that the identification rate of the two algorithms can be changed regarding the needs of the user and the selection of the appropriate threshold criteria.

With the used criteria in the NATALI software (with a confidence for the output retrieval higher than 0.9), the algorithm returns only the retrievals that pass a number of quality checks, so as to make sure (as possible) that the output is trustworthy. As such, the percentages of 'unknowns' are not because NATALI cannot identify the aerosol type, but due to large uncertainties of the optical parameters and the calculated uncertainties. However, these 'unknowns' layers were typed (89% and 11% stayed untyped), when the minimum acceptance confidence were chosen differently (with a confidence higher than 0.7). Specifically, from the 28 'unknowns' layers, 26 were typed when the minimum acceptance confidence is chosen lower, 2 remained 'unknowns'. From these 26 cases 11 (40%) give the same aerosol type with EARLINET Mahalanobis distance typing algorithm. A new plot is added in the revised version of the manuscript, explaining each classification.

From the mismatches: 7 cases were attributed as Dust from EARLINET Mahalanobis distance typing algorithm (Hysplit back-trajectories don't agree) and identified from NATALI either as Continental or Smoke. Higher values of Lidar ratio at 532nm are allowed for EARLINET Mahalanobis distance typing algorithm in identifying dust particles (48-62sr), considering all dust-like aerosol types as one category. Therefore, a number of Dust recognition (from EMD) are attributed to Smoke particles or CleanContinental particles in the NATALI output. CleanContinental(NATALI) and Maritime (EARLINET

Mahalanobis distance typing algorithm) don't agree most of the times. Also, a Dust case (typed by NATALI) is attributed as Smoke from the EARLINET Mahalanobis distance typing algorithm (back-trajectories indicate air masses from fires spots).

Nevertheless, we choose to show the comparison, holding the retrievals that pass a higher percentage of quality checks.

Also, when the EARLINET Mahalanobis distance typing algorithm is forced to categorize a layer, only when the normalized probability is above 60% (instead of 50% that used in the comparison), the number of untyped layers increased (20 untyped cases, 81% classification of the total number of cases). These layers are the same that failed to be typed with the higher percentage of quality checks in NATALI. Nevertheless, we show the comparison for the criteria chosen also in the study of Papagiannopoulos et al., (2018).

Once again, we conclude that the identification rate of the two algorithms can be changed regarding the needs of the user and the selection of the appropriate threshold criteria.

Therefore the following paragraph has been added in the revised mansucript/

"NATALI classified 82 of the total 110 layers (75% of the total number of cases) and 28 were flagged as Unknown layers. These missed cases are based mostly on retrievals with larger uncertainties of the optical parameters and the calculated uncertainties, that not allowed ANNs to return any aerosol type. However, these 'unknowns' layers were typed (93% and 7% stayed untyped), when the minimum acceptance confidence was chosen lower (i.e., confidence higher than 0.7). Nevertheless, we choose to show the comparison, holding the retrievals that pass a higher percentage of quality checks. The EARLINET Mahalanobis distance-based typing algorithm classified 98 layers (89%) and 12 layers were not assigned to any cluster (the Mahalanobis distance values larger than 4), showing a higher identification rate. However, when we set the second of the screening criteria (i.e., the normalized probability of the aerosol class) to be higher than 50%, the number of untyped layers increased. We conclude that, the identification rate can be changed regarding the needs of the user and the selection of the appropriate threshold criteria."

*46) Line 5 page 9 the dust cases expressed as percentages whereas these are absolute numbers. why the difference? choosing one or the other (percentages, perhaps) would make it easier for readers to put the performance on the different type into a uniform context.*

The reviewer is right. Text has modified in the revised version, according to the reviewer's suggestion:

"The agreement is remarkable for the PollutedSmoke cases, with occurrence ratio of 43% and 47% for NATALI and EARLINET Mahalanobis distance-based typing algorithm, respectively, whilst the CleanContinental cases showed a difference with 18% and 24% for NATALI and EARLINET Mahalanobis distance-based typing algorithm, respectively."

*47) Line 10 page 9 why is this not also true for NATALI? (I'd think the definition of the aerosol classes would be "of paramount importance" for every aerosol classification scheme)*

The reviewer is right. The text has been modified to:

"Additional to the ability of the two automated algorithms to attribute the correct classification to the aerosol layers, also the uncertainties of the input measurements have to be taken into account. Both algorithms are unable to return aerosol type for not perfect calibrated input data, i.e., when one or more intensive optical properties are not perfect."

*48) Line 14 page 9 and perhaps because the dust classes are defined similarly for the two different schemes???*

This is not true. As we can see from Table 3, the mean aerosol optical properties of the reference aerosol types used on the two automated algorithms, are quite different for the dust category. Higher values of Lidar ratio at 532nm are allowed for EARLINET Mahalanobis distance typing algorithm in identifying dust particles (48-62 sr), considering all dust-like aerosol types as one category. Therefore, a number of Dust recognition are attributed to PollutedSmoke particles (6 cases) or CleanContinental particles (1 case) in the NATALI output.

*49) Line 22 page 9 this is a nice summary that should be introduced much earlier in the discussion*

The reviewer is right. The table is provided earlier in the manuscript (In Section 3.3) and the following sentence is added in the methodology part:

"Table 2 lists the classified aerosol types of the above study and Table 3 lists the mean aerosol optical properties of the reference aerosol types defined by the two algorithms."

*50) Line 24 page 9 please explain how the model parameters are converted into synthetic lidar measurements. e.g., were realistic amounts of random noise added? how about systematic errors, like lidar calibration? were Ångström exponents used to convert 350 nm and 550 nm model parameters into 355 and 532 lidar measurements?*

The ability of the ANNs to retrieve the aerosol type depends strongly on the physical content and the uncertainty of the optical inputs as well as on the structure of the ANN and the training process, including the extent of the data set used for this purpose. To create a consistent picture of the aerosol types, an aerosol model representing the optical properties of different aerosol was developed. The relative errors considered here are 50 % for the extinction, 20 % for the backscattering and 30 % for the depolarization. Note that these values were assumed to be inclusive to mimic high-precision but also moderate-precision retrieved parameters. The synthetic database was developed using the aerosol model built for 350, 550, and 1000 nm sounding wavelengths, based on the 61 wavelengths of OPAC (Optical Properties of Aerosols and Clouds) software package (Hess et al., 1998), for which the microphysical characteristics of the aerosols are available from GADS (Global Aerosol Data Set). These wavelengths are then re-scaled to the usual lidar wavelengths (i.e., 355, 532 and 1064 nm), that are used as input to NATALI, using an average Ångström exponent equal to one.

*51) Line 25 page 9 that the EARLINET Mahalanobis algorithm failed to make a typing decision 39% of the time tells me that the underlying models are very different. so different, in fact, that I question the value of comparing the outputs of the two different classification schemes.*

See our response to comment 45.

*52) Line 28page 9 the authors need to explain how the table values are computed; that is, what quantities are used to compute the percentages shown? since each column adds up to 100%, and since the input data comes from the NATALI models (and hence the "true" type for each sample is known), should we assume that the denominator in each column is the number of NATALI samples for a given type, and the numerator is the number of Papagiannopoulos classifications? I'd much prefer to read the authors' unambiguous explanation, rather than relying on my own assumptions about how the values were derived.*

The reviewer is right and the text has been modified to:

"Each percentage is defined as follows. The denominator in each column is the number of NATALI samples for a given type, and the numerator is the number of EMD classifications for the same sample."

*53) Line 3 page 10 since NATALI failed to classify 36 of 116 cases, this "very good agreement" applies only to ~2/3 of the full data set. the overall agreement between the two methods is best characterized as poor.*

NATALI classified 82 of the total 110 layers (75% of the total number of cases) and 28 were flagged as Unknown layers. These missed cases are based mostly on retrievals with larger uncertainties of the optical parameters and the calculated uncertainties, that not allowed ANNs to return any aerosol type.

Indeed, the agreement can be characterized as a poor one, as it showed here, but we should keep in mind that with the used criteria in the NATALI software, the algorithm returns only the retrievals that pass a number of quality checks, so as to make sure (as possible) that the output is trustworthy. See our detailed response to comment 45.

The revised text is modified accordingly.

*54) Line 4 page 10 again, it's essential that the authors explain how the percentages shown in this figure are calculated. in particular, please explain how the percentages in figure 5 relate to the sample sizes shown in figure 3.*

The reviewer is right. Figure percentages are replaced by numbers, so to be consistent with the sample sizes of Figure 3. The information now is more straightforward and a paragraph is added explaining the numbers.

*55) Figure 1 since the Papagiannopoulos algorithm uses AE355/1064, I would expect that quantity to be plotted here. it's a bit misleading to show 14 possibilities here when the NATALI algorithm in this study is constrained to identifying only 4 types.*
*please add text explaining why the probabilities do not add up to one.*

The reviewer is right. The  BAE 355/1064 is provided in the plot of the revised version. Also, the labels of the NATALI output were changed to the low resolution output. The text explaining the probabilities also added  and given below:
"The plotted probabilities are assigned to each calculated distance and are different from the normalized ones that used as a screening criteria (and add up to 100)." See also comment on 28.

*56) it's a bit misleading to show 14 possibilities here when the NATALI algorithm in this study is constrained to identifying only 4 types*

The reviewer is right. The labeled categories are corrected in the plot. But, as explained above, the NATALI low resolution typing, consists of 5 types (+ the Unknown category and the N/A). See also, Table 2.

*57) Figure 4 since each column adds to 100%, it would be very helpful (essential, even?) to know the number of samples in each column. are the samples uniformly distributed among the NATALI aerosol types*

The number of samples in each column are added to the text.
"Fair agreement is found for the NATALI synthetic datasets of Dust and PollutedSmoke categories. In particular, for Dust reached agreement of 83.7% (100166 typed over 119650). This result is very good considering that up to 30% of the aerosol class contains mixtures. For the PollutedSmoke, the majority of the synthetic data, corresponding to 48180 instances from a total typed of 82041 (58.7%) is classified as PollutedSmoke. For the CleanContinental type, the agreement is poor (15%), as only 7605 over 52058 were correctly classified and the majority of the instances (36552) is typed as Dust (70%). This Dust-CleanContinental mismatch is confirmed also by the 11.8% of the NATALI reference dust cases that are typed as Clean Continental by the EARLINET Mahalanobis distance-based typing method. Furthermore, 7901 of the instances (15%) are classified as PollutedSmoke. Finally, the agreement for Maritime is 23% of the cases (1319 from a total of 5658 cases), indicating the different class definitions as well as the big spread of the NATALI maritime model that causes the EARLINET Mahalanobis distance-based typing algorithm to misclassify those data."

There are some typo that are corrected accordingly:

Line 15 page 4: 'at 355 nm' is added

Line 20 page 4: 'discriminate' replace 'detect'

Line 2 page 6: 'available' is added

Line 3 page 11: 'reasonably close' replace 'high enough'

Line 34 page 11: 'Furthermore' is corrected

The following references have been added in the revised manuscript

Amiridis, V., Marinou, E., Tsekeri, A., Wandinger, U., Schwarz, A., Giannakaki, E., Mamouri, R., Kokkalis, P., Binietoglou, I., Solomos, S., Herekakis, T., Kazadzis, S., Gerasopoulos, E., Proestakis, E., Kottas, M., Balis, D., Papayannis, A., Kontoes, C., Kourtidis, K., Papagiannopoulos, N., Mona, L., Pappalardo, G., Le Rille, O., and Ansmann, A.: LIVAS: a 3-D multi-wavelength aerosol/cloud database based on CALIPSO and EARLINET, Atmospheric Chemistry and Physics, 15, 7127–7153, doi:10.5194/acp-15-7127-2015, http://www.atmos-chem-phys.net/15/7127/2015/, 2015.

D'Amico, G., Amodeo, A., Baars, H., Binietoglou, I., Freudenthaler, V., Mattis, I., Wandinger, U., and Pappalardo, G.: EARLINET Single Calculus Chain – overview on methodology and strategy, Atmos. Meas. Tech., 8, 4891–4916, https://doi.org/10.5194/amt-8-4891-2015, 2015.

Giannakaki, E., Pfüller, A., Korhonen, K., Mielonen, T., Laakso, L., Vakkari, V., Baars, H., Engelmann, R., Beukes, J. P., Van Zyl, P. G., Josipovic, M., Tiitta, P., Chiloane, K., Piketh, S., Lihavainen, H., Lehtinen, K. E. J., and Komppula, M.: One year of Raman lidar observations of free-tropospheric aerosol layers over South Africa, Atmospheric Chemistry and Physics, 15, 5429–5442, doi:10.5194/acp-15- 5429-2015, 2015.

Granados-Muñoz, M.J., Navas-Guzmán, F., Luis Guerrero-Rascado, J., Antonio Bravo-Aranda, J., Binietoglou, I., Nepomuceno Pereira, S., Basart, S., Baldasano, J.M., Belegante, L., Chaikovsky, A., Comerón, A., D'Amico, G., Dubovik, O., Ilic, L., Kokkalis, P., Muñoz-Porcar, C., Nickovic, S., Nicolae, D., José Olmo, F., Papayannis, A., Pappalardo, G., Rodríguez, A., Schepanski, K., Sicard, M., Vukovic, A., Wandinger, U., Dulac, F., Alados-Arboledas, L. Profiling of aerosol microphysical properties at several EARLINET/AERONET sites during the July 2012 ChArMEx/EMEP campaign Atmospheric Chemistry and Physics, 16 (11), pp. 7043-7066., 2016.
Groß, S., Esselborn, M., Weinzierl, B., Wirth, M., Fix, A., and Petzold, A.: Aerosol classification by airborne high spectral resolution lidar observations, Atmospheric Chemistry and Physics, 13, 2487–2505, doi:10.5194/acp-13-2487-2013, http://www.atmos-chem-phys.net/13/2487/2013/, 2013.

Haarig, M., Ansmann, A., Baars, H., Jimenez, C., Veselovskii, I., Engelmann, R., and Althausen, D.: Depolarization and lidar ratios at 355, 532, and 1064 nm and microphysical properties of aged tropospheric and stratospheric Canadian wildfire smoke, Atmos. Chem. Phys., 18, 11847-11861, https://doi.org/10.5194/acp-18-11847-2018, 2018.

Hess, M., Koepke, P., and Schult, I.: Optical properties of aerosols and clouds: The software package OPAC, B. Am. Meteorol. Soc., 79, 831–844, https://doi.org/10.1175/1520-0477(1998)0792.0.CO;2, 1998.

Mona, L., Amodeo, A., D'Amico, G., Giunta, A., Madonna, F., and Pappalardo, G.: Multi-wavelength Raman lidar observations of the Eyjafjallajökull volcanic cloud over Potenza, southern Italy, Atmos. Chem. Phys., 12, 2229-2244, https://doi.org/10.5194/acp-12-2229-2012, 2012.

Mona L., Kahn R., Papagiannopoulos N., Pappalardo G., Holzer-Popp T.: Aerosol typing–key information from aerosol satellite measurements, European Aerosol Conference, Milan (Italy) September 6-11, https://geko.promeeting.it/abstract/DEF/Poster/2AAP_P094.pdf, 2015.

Schwarz, A.: Aerosol typing over Europe and its benefits for the CALIPSO and EarthCARE missions – statistical analysis based on multiwavelength aerosol lidar measurements from ground-based EARLINET stations and comparison to spaceborne CALIPSO data, PhD thesis, University of Leipzig, 2016.

Sicard, M., Guerrero-Rascado, J. L., Navas-Guzmán, F., Preißler, J., Molero, F., Tomás, S., Bravo-Aranda, J. A., Comerón, A., Rocadenbosch, F., Wagner, F., Pujadas, M., and Alados- Arboledas, L.: Monitoring of the Eyjafjallajökull volcanic aerosol plume over the Iberian Peninsula by means of four EARLINET lidar stations, Atmos. Chem. Phys., 12, 3115–3130, https://doi.org/10.5194/acp-12-3115-2012, 2012.

Shin, S.-K., Tesche, M., Kim, K., Kezoudi, M., Tatarov, B., Müller, D., and Noh, Y.: On the spectral depolarisation and lidar ratio of mineral dust provided in the AERONET version 3 inversion product, Atmos. Chem. Phys., 18, 12735-12746, https://doi.org/10.5194/acp-18-12735-2018, 2018

Yorks J.E., Palm S.P., McGill, M.J., Hlavka D.L., Hart, W.D., Selmer P.A., and Nowottnick E.,: CATS Algorithm Theoretical Basis Document, Level 1 and Level 2 Data Products, https://cats.gsfc.nasa.gov/media/docs/CATS_ATBD.pdf, 2016.

---

## Author Comment (AC2) · 22 Apr 2019

We would like to thank the reviewer for his/her fruitful comments that led to the improvement of the manuscript. In the following, answers to comments are reported just below each related comment. When needed, the part of the manuscript we modified or added to the old version, is reported. In the revised version of the paper all comments have been extensively taken into account discussing more aspects of the comparison between the two typing techniques (motivation, agreement and disagreement of the methods). Some text has been added about this and it is reported below into the replies to specific comments.

*1) Title 'Thessaloniki'*

We keep the more generic title, since the concept of the paper is not to discuss explicitly the aerosol types observed in Thessaloniki, but is more about discussing an approach for evaluating the typing methods.

*2) Line 13-14 ten : marine, dust, polluted continental / smoke, clean continental, polluted dust, elevated smoke, dusty marine, PSC aerosol, volcanic ash, sulfate/other*

We thank the reviewer for his/her correction. The text has been modified to:

"The CALIPSO mission uses a decision-tree based on lidar profiles and external data (Omar et al., 2009) in order to classify the aerosol load in ten aerosol subtypes, i.e., marine, dust, polluted continental / smoke, clean continental, polluted dust, elevated smoke, dusty marine, PSC aerosol, volcanic ash, sulfate/other."

*2) Line 6 page 3 You can also provide literature for other European countries (eg Greece)*

The text has been modified to: "Simultaneous observations of desert dust and ash particles were made during the Eyjafjallajokull volcanic eruption in 2010 and the methodology for the type discrimination was presented by Papayannis et al. (2012); Mona et al. (2012); and Pappalardo et al. (2013)."

*3) Line 10 page 3 in the abstract you are refer to: dust, maritime, polluted smoke and clean continental. Please be consistent.*

The reviewer is right. The text has been modified to:
"The two automatic aerosol classification methods and the methodology used to characterize the layers in four basic aerosol types (i.e., Dust, PollutedSmoke, Maritime, CleanContinental) are presented in Sect. 3".

*4) Line 24 page 3 Please provide a more recent paper that actually describe the recent status of THELISYS*

The text has been modified to:
"A detailed description of THELISYS can be found in Siomos et al. (2018a) and Siomos (2018c)."

*5) Line 2 page 4 Here you should explain the two different options of the aerosol typing algorithms (with and without δ)*

Both automated algorithms can ingest depolarization information, however in our study we did not consider this possibility. The reason for this is the depolarization ratio unavailability. The next sentence text is changed to:

"The input module requires optical properties profiles as those measured by EARLINET stations, namely the aerosol extinction coefficient and the aerosol backscatter coefficient profile. Optionally, the linear particle depolarization profile at 532 nm can be provided, so as to allow a better classification and to increase the number of classified aerosols types."

A complete description of the two different options of the aerosol typing algorithms (with and without δ), is given in Section 3.1 and Section 3.2. See below the paragraphs modified in the revised version:

"Depending on the availability of the particle linear depolarization ratio and the quality of the provided lidar profiles, the derived typing can be either of high resolution (AH), or low resolution with depolarization (AL) or low resolution without depolarization (BL). Pure aerosols categories, and even mixtures of three aerosols types can be obtained from the NATALI algorithm. In the high resolution typing, 14 aerosol types can be distinguished (i.e., Continental, ContinentalPolluted, Dust, Maritime/CC, Smoke, Volcanic, Coastal, CoastalPolluted, ContinentalDust, ContinentalSmoke, DustPolluted, MaritimeMineral, MixedDust and MixedSmoke) when the quality of the provided optical products is high enough. In the low resolution typing, 6 predominant aerosol types can be provided but with high uncertainty (i.e, Continental, Continental polluted, Smoke, Dust, Maritime and Volcanic). The low resolution typing provides 5 predominant aerosol types (Dust, ContinentalPolluted, Smoke, Continental, Maritime) either pure or mixed, when the depolarization information is not provided."

"The algorithm applies the Mahalanobis distance classifier (Mahalanobis, 1936) to classify observations into maximum 8 (Dust, Volcanic, Mixed Dust, Polluted Dust, Clean Continental, Mixed Marine, Polluted Continental, Smoke) and minimum 4 (Dust, Maritime, PollutedSmoke, CleanContinental) aerosol classes, considering the needs of each user and the availability of the intensive properties."

Also, the Table 2 added in the revised version, contains all the necessary information, describing both the high and low resolution aerosol typing provided by each algorithm and the low resolution that is used in our study.

*6) Line 20 page 4 What about the particle depol?*

The reviewer is right, therefore the following sentence is added in the revised manuscript:
"Optionally, the linear particle depolarization profile at 532 nm can be provided, so as to allow a better classification and to increase the number of classified aerosols types."

*7) Line 25page 4 In the previous paragraph you mention that only intensive optical properties are used, but here the backscatter is used for the identification of the layers*

Yes, the reviewer is right. The intensive optical properties are used for the typing, but the backscatter at 1064 nm is used only for the identification of the aerosol layers.

*8) Line 6 page 5 How is high enough defined?*

We can talk about high quality of lidar products, when the values of the intensive optical parameters are between acceptable limits (see Table 1) and the relative error of each intensive optical parameter is lower than 50%.

*9) Line 7 page 5 you mean low resolution with depol information*

Yes, the reviewer is right. The low resolution (AL) typing corresponds to the low resolution with depolarization information.

The text is modified accordingly: In the low resolution typing (AL), 6 predominant aerosol types can be provided but with high uncertainty (i.e, Continental, ContinentalPolluted, Smoke, Dust, Maritime and Volcanic). The low resolution typing (BL) provides 5 predominant aerosol types (Dust, ContinentalPolluted, Smoke, Continental, Maritime) either pure or mixed, when the depolarization information is not provided.

*10) Line 8-9 page 5 Here the types are 5, in the comparison you refer to 4 aerosol types…*

The reviewer is right. The methodology section is changed accordingly:

"The NATALI typing was performed in the low resolution typing configuration (5 predominant aerosol types - Dust, Smoke, Continental Polluted, Continental and Maritime) since particle linear depolarization ratio measurements for Thessaloniki were not available for the study period. In what follows, we merged the output types from NATALI that tend to reflect the same aerosol characteristics, and hence we evaluate the corresponding effects on the prediction rate of the algorithms. Thus, the smoke and the polluted continental categories were grouped into the more generic type of small particles with high lidar ratio values. The selection of four main aerosol classes stems from the availability of intensive properties, the difficulty in deriving a confident classification without particle linear depolarization ratio and the difficulty in discriminating polluted continental and smoke particles that reveal the same type characteristics. Regardless, the aerosol classes describe the major aerosol components. The identified layer boundaries from NATALI are used as input in the EARLINET Mahalanobis distance-based typing algorithm. Considering the aforementioned typing merging, the EARLINET Mahalanobis distance-based typing algorithm was set to classify observations into 4 aerosol classes: CleanContinental, Dust, Maritime and PollutedSmoke."

Also, the following sentence is added in the last paragraph of the methodology section:

"The idea here is to compromise: i) the resolution (low) of the automatic classification owing to the availability of the optical properties (i.e., 3+2 lidar configuration), and, ii) the type definition, which

does include the wide spectrum of the aerosol types provided by the two automated typing techniques."

Table 2 is added and lists the classified aerosol types of the above study.

The paragraph is reworded and reads as follows:

"Finally, the assessment of the predictive performance of the algorithm was tested on a testing dataset. For this purpose, EARLINET data collected during the ACTRIS Summer 2012 intensive measurements (Sicard et al., 2015; Granados-Muñoz et al., 2016b) were chosen to test the automatic typing algorithm. The testing dataset comprised of 47 layers, 21 of which yielded depolarization ratio values. The performance of the algorithm was checked for each of the grouping classes (i.e., 8,7,6,5,4) and the predictive accuracy of the algorithm increased up to 90% when the aerosol classes that tend to reflect the same optical properties values were combined into 4 (Dust, Maritime, Polluted Continental + Smoke, Clean Continental) without providing the information of the depolarization. The study concluded that the fewer aerosol classes (i.e., 4, 5, 6 classes) could provide a successful prediction accuracy, even without depolarization values, but, nonetheless, a coarser and less insightful classification."

*12) Line 2 page 6 I don't understand. In this study? Increase to 90% compared to what? In general? please provide reference*

Papagiannopoulos et al. (2018), checked the performance of the algorithm for each of the grouping classes and the predictive accuracy of the algorithm increased up to 90% (See Figure 10 in Papagiannopoulos et al., 2018) when the aerosol classes that tend to reflect the same optical properties were combined into 4 without providing the information of the depolarization (compared to 80% when depolarization ratio information was included). The study, also, showed that without depolarization ratio information, the accuracy of the model increases with decreasing number of classes, providing however a coarser classification. Instead, the training of the classification with depolarization measurements enhances the predictability strength and can provide finer aerosol classification (for 8 classes).

*13) Line 3-5 page 6 So, you have 90% accuracy without using the depol information?*

The reviewer is right. In general, the particle linear depolarization ratio increases the ability for correctly predicting the aerosol type. However, Papagiannopoulos et al. (2018) showed that (Figure 10), without depolarization ratio information, the accuracy of the model increases with decreasing number of classes (90% for the 4 classes), providing however a coarser classification.

*14) Line 3-5 page 6 I understand that you have an accuracy of 90% for 4 classes. But fewer aerosol classes (I suppose fewer than 8..).*

The reviewer is right. Papagiannopoulos et al. (2018) analyzed the predictive accuracy of the algorithm when compared to manually analyzed data for the different aerosol classes in both the cases in which

the depolarization information is available, for 8, 7, 6, 5 and 4 classes. This highlights that the typing in multiple classes and the typing accuracy are two conflicting aspects. The number of classes as well as the typing accuracy depends on the specific needs. This could be an approach for the specific user to select the appropriate balance each specific application. Another possibility is to find a compromise between degrading accuracy and gaining insight into the aerosol type.

*15) Line 16 page 6 I suppose that you merged Smoke with Continental polluted. Please clarify this in the text*

The reviewer is right. This type combination was not clearly stated clearly in the text. The selection of four aerosol classes stems from to the difficulty to distinguish aerosol types of aerosols that reflect the same aerosol characteristics. Besides, the particle depolarization ratio is not available for this study, which constitutes a powerful aerosol type discriminator. Therefore, we merged Smoke with Continental polluted. Following this comment, we added a table (Table 2) with the aerosol classes for both the High and Low resolution (without depolarization) mode for both automatic typing techniques and the aerosol types used in this study. Additionally, Papagiannopoulos et al. (2018) showed that in the testing phase of the algorithm, that the Dust classification showed a high success rate, whilst the aerosol types that performed worse were the smoke and polluted continental aerosol. However, when these two categories were combined into a single aerosol class, the correct prediction increased.

So, the following paragraph is added in the revised version of the text:

"In what follows, we merged the output types from NATALI that tend to reflect the same aerosol characteristics, and hence we evaluate the corresponding effects on the prediction rate of the algorithms. Thus, the smoke and the polluted continental categories were grouped into the more generic type of small particles with high lidar ratio values. The selection of four main aerosol classes stems from the availability of intensive properties, the difficulty in deriving a confident classification without particle linear depolarization ratio and the difficulty in discriminating polluted continental and smoke particles that reveal the same type characteristics. Regardless, the aerosol classes describe the major aerosol components. The identified layer boundaries from NATALI are used as input in the EARLINET Mahalanobis distance-based typing algorithm. Considering the aforementioned typing merging, the EARLINET Mahalanobis distance-based typing algorithm was set to classify observations into 4 aerosol classes: CleanContinental, Dust, Maritime and PollutedSmoke."

*16) Line 21-28 page 6 Maybe a simple sketch is useful here.*

We excluded this paragraph and instead section 3.1 was changed to the following one, in order to provide all the necessary information.

"The identification of the most probable aerosol type is then made through a voting procedure, using the results from the three ANNs interrogated. Over 50000 aerosol synthetic data have been used to train the ANN and identify the better ANNs to classify the aerosols type from multiwavelength lidar data. The capability of ANNs to resolve the overlapping clusters of the intensive optical parameters is used on NATALI algorithm. The answer is selected based on a statistical approach. The selected types

of ANNs classify the aerosols based on the response with high i) confidence (i.e. the probability of having one of the aerosol types) and ii) stability over the uncertainty range (i.e., the percentage of agreement for values between error limits). Therefore, answers with low confidence are filtered out and NATALI returns the 'Unknown' type. In this study, we select the confidence level for the output retrievals higher than 0.9 (minimum accepted confidence) and the minimum agreement threshold as default (i.e., 0.25), so as to make sure (as possible) that the output typing is trustworthy."

*17) Line 31 page 6 you mean clean continental?*

We thank the reviewer for his/her comment. The text has been modified to:
"and small particles with medium lidar ratios (i.e., clean continental particles, with mean lidar ratios, for 355 and 532 nm, 50 ± 8 and 41 ± 6 sr, respectively)."

*18) Line 31 page 6 I think here is necessary to give number what is low / medium and high lidar ratios and what is large and small particles*

Type specific values are added to the text.

"Consequently, the lidar classification scheme consists of the main classes: (i) large particles with medium lidar ratios (i.e., dust-like particles, with mean lidar ratios of 58 ± 7 and 48 ± 55 sr for 355 nm and 532 nm, respectively, Groß et al. (2013)), (ii) large particles with low lidar ratios (i.e., maritime particles, with mean lidar ratio at 355nm and 532nm of 18 ± 4 and 18 ± 2 sr, respectively, Groß et al. (2011)), (iii) small particles with high lidar ratios (i.e., pollution and/or smoke particles, the smoke mean lidar ratio values present values of 81±16 and 78±11 sr for 355 and 532 nm, respectively – and the polluted continental values succeed with 69±12 and 63± 13 sr for 355 and 532 nm, respectively, Amiridis et al., 2009; Baars et al., 2012) and (iv) small particles with medium lidar ratios (i.e., clean continental particles, with mean lidar ratios, for 355 and 532 nm, 50 ± 8 and 41 ± 6 sr, respectively). Generally, desert dust layers have optical properties that are considerably different from the other types, thus they are easily identified. Their big size leads to low Ångström exponent values and the reported lidar ratio at 355nm ranges from 47 to 58 sr for Thessaloniki (Siomos et. al., 2018). PollutedSmoke particles, are also easily identified, as they are highly absorbing particles, with high lidar ratio values. CleanContinental categorization is not completely straightforward, because the continental particles can be attributed to different subcategories (i.e., local, continental polluted or mixtures). In general, the CleanContinental cases are typically elevated layers, i.e. layers not related to the local atmospheric boundary layer where the pollution and anthropogenic contribution would mean more absorbing particles and therefore labeled as PollutedSmoke aerosol. Continental particles present low lidar ratio values, (i.e., 20–40 sr) and values of Ångström exponents around 1.0–2.5. The highest values appear during summer period in Thessaloniki (Siomos et al., 2018)."

*19) Line 1 page 7 I would call this category Mixed Dust and not Dust*

The reviewer is right. Dust category can be either pure dust, or mixed dust, or polluted, or volcanic. But, as mixed Dust and Polluted Dust are different definitions for NATALI typing scheme, we keep label

the category with the generic 'Dust'. This is specified in Table 2, which lists the classified aerosol types of the above study.

*20) Line 2 page 7 So, is there any layer Clean Continental in the urban city of Thessaloniki?*

Clean continental type is defined in the CALIPSO scheme background aerosol and as a consequence, deemed not to be influenced by urban pollution. However, these conditions are probably not realistic for the European continent. So, this category was revised for the applicability to Europe in the two typing schemes, as the clean continental aerosol over Europe is a mixture of anthropogenic pollution with particles from natural sources, but with a predominance of no-anthropogenic aerosol resulting a low lidar ratio observed value (Papagiannopoulus et al., 2018).

As reported in the text many layers were typed as Clean Continental over Thessaloniki and these are typically elevated layers (90% of the detected CleanContinental cases are found above 2km). This result is compliant with the CALIPSO scheme at global level and with results about Thessaloniki site characterization done by Siomos et. al (2018). In their paper Siomos et al. (2018b), used data from a double monochromator Brewer spectrophotometer and a sunphotometer in order to classify aerosol cases during the period 2007-2017 in Thessaloniki, in the following categories: Water soluble, Black Carbon, Dust, Sea Salt and mixed. They found that the pure Water Soluble category (which can be related to the Clean Continental consisting of mainly of water soluble particles) correspond to 29.1% of the cases, which is in fair agreement with results reported in Fig.3.

We added a sentence about this in the revised version of the paper:

"CleanContinental categorization is not completely straightforward, because the continental particles can be attributed to different subcategories (i.e., local, continental polluted or mixtures). In general, the CleanContinental cases are typically elevated layers, i.e. layers not related to the local atmospheric boundary layer where the pollution and anthropogenic contribution would mean more absorbing particles and therefore labeled as PollutedSmoke aerosol. Continental particles present low lidar ratio values, (i.e., 20–40 sr) and values of Ångström exponents around 1.0–2.5. The highest values appear during summer period in Thessaloniki (Siomos et al., 2018)."

*21) Line 3 page 7 I suppose that this category is pure maritime+mixed maritime*

True. For making clearer this point, we added a table (Table 2) describing each aerosol category.

*22) Line 7 page 7 even without depolarization ratio?and even without back-trajectory analysis?*

Usually dust can be identified, since the optical properties of dust particles are quite different from the other three classes. Even though we do not consider depolarization ratio information, the size information (i.e., Angstrom exponent) and physical and chemical properties (i.e., lidar ratio) are usually sufficient to discriminate Dust. Nicolae et al. (2018) and Papagiannopoulos et al. (2018) have demonstrated this possibility. Backward trajectory analysis is a well-established tool for the source identification, however automated classification algorithms for lidars, nowadays, can operate stand-alone and provide robust results (e.g., Omar et al., 2009; Burton et al., 2012; Nicolae et al., 2018;

Papagiannopoulos et al., 2018). Indeed, support from back-trajectory analysis and/or model simulations are necessary for an independent evaluation of the stand-alone typing algorithms and this is the reason we demonstrate in our paper certain case studies (section 4.1, 4.2, 4.3)

*23) Line 7 page 7 Please provide numbers*

Groß et al. (2011) for example reported values of 0.06 ± 0.21 and Tesche et al. (2009b) reported values of 0.19 ± 0.20. But, it should be pointed out that different values reported can be related to the different transportation paths of the load.

*24) Line 10 page 7 Small? I suppose you are referring to aged smoke or polluted continental. Fresh smoke is not small*

Yes, the reviewer is right. Smoke particles yield a wide range of optical properties. The transportation path, the aging, the burning material, the hygroscopic growth, and the height of injection are some of the parameters that can affect the observed optical properties. For instance, smoke layers observed by EARLINET systems over Europe in the summer of 2017 indicated high depolarization ratio (e.g., Haarig et al., 2018). The mechanisms for this are still yet to be understood. Here, we merged Smoke with Continental polluted, and the characteristics correspond to the merged class. We would like, also, to comment that we had to i) compromise the resolution of the automatic classification owing to the availability of the optical properties (i.e., 3+2 lidar configuration), and, ii) the type definition, which does include the wide spectrum of the aerosol types as stated above.

*25) Line 32 page 7 Please correct the label of Natali in Figure 1, it should have run for 5 (merged to 4) classes*

Following this comment, the plot is changed and the NATALI labels, now, are 5 (+ 2 categories of 'Unknown' and 'N/A' typing), corresponding to the categories of the low resolution typing.

*26) Line 1 page 8 Please be consistent: use either Layer A, B or Layer 1, 2*

The plot and the text is changed accordingly, using the labels of Layer 1, 2.

*27) Line 2 page 8 'and Angstrom exponent'*

Following this comment, the text is changed to: 'The stability of the lidar ratio and Ångström exponent values could be considered as an indicator of homogeneity and small variability of the aerosol type within the layer.'

*28) Line 12 page 8 please present the fire spots in the trajectory already provided in Figure 2. Also give reference for the fires.*

The fire spots are added in Figure 2. These values are in accordance with the typical biomass burning values observed over Thessaloniki. Giannakaki et al. (2010) report an annual mean lidar ratio at 355nm of 69 ± 17 sr and a mean BAE at 355-532nm of 1.7 ± 0.7, while Siomos et al. (2018a) found the lidar ratio at 355nm ranging from 51 to 73 sr for biomass burning events.

The paragraph is reworded and reads as follows:

"These values are in accordance with the typical biomass burning values observed over Thessaloniki. Giannakaki et al. (2010) reported an annual mean lidar ratio at 355nm of 69 ± 17 sr and a mean BAE at 355-532nm of 1.7 ± 0.7, while Siomos et al. (2018a) found lidar ratio at 355nm ranging from 51 to 73 sr for biomass burning events."

*29) Line 23 page 8 I would like to see, the application of the algorithms not only to pure dust and bb cases. To classify these pure cases is relative easy and there is not a need for aerosol classification algorithm. The authors should provide additional case studies with mixed aerosol types, or cases with complex aerosol structure and check the reliability of the algorithms.*

The reviewer is right. A new plot and a paragraph (Section 4.3) is added in the revised version. The typing scheme selected and added is a more complex one and offers the opportunity to check the reliability of the algorithms in conditions where different aerosol types at different heights are observed.

*30) Line 33 page 8 It is difficult to believe that the second aerosol type observed in Thessaloniki is clean continental. Have you also compared these results with satellite/model products? Clean continental is defined as 'background like' aerosols with a LR of about 30-35sr. I would speculate that the observed in this study layers would be small, non so absorbing aerosol.*

See also our response to comment 20. Typically, the clean continental type presents the mixture of anthropogenic aerosols with natural sources. For instance, the manual typing made by Schwarz et. al. (2016) in the framework of EARLINET, assigns an aerosol layer as clean continental when the aerosol concentration is low by means of optical depth (the low values of lidar ratio). Based on this definition, the label "clean" can be taken literally. However, the automatic typing procedures do not take into account any extensive parameter, hence not "clean" aerosol layers might be classified as clean continental.

*31 ) Line 3 page 9 We should talk for agreement only if the same layer is attributed to the same class. In such a figure we only new the statistical information of the occurrence of the layers, not if the classification is made right.*

The reviewer is right. The paragraph is reworded and reads as follows:

"In particular, the agreement is reasonably close for the desert dust cases (10% and 17% for NATALI and EARLINET Mahalanobis distance-based typing algorithm, respectively), nevertheless, it becomes evident that the particle linear depolarization ratio could increase the ability for correctly predicting dust particles."

*32) Line 24 page 9 Please explain possible reasons for that*

After the bug in the processing, the scores have changed to the following ones and the text has been modified to:
"For PollutedSmoke and CleanContinental the accuracy reached 88% and 65% respectively."

In order to address the possible reasons of disagreement, we added the following paragraph in section 4.4 of the revised version of the manuscript and a summary table (Table 3) which provides mean aerosol optical properties of the reference aerosol types used on the two automated algorithms.

[revised manuscript text omitted]

*33) Line 12 page 10 case studies*

The text has been modified to:
"The prediction of the automatic classification methods in the three case studies showed consistent results when compared against manually classified EARLINET data."

*34) Figure 1*

*S stands for Pure smoke or general for the category: polluted continental + smoke?*

The reviewer is right. The PS label is added to the plot, that corresponds to the category: polluted continental + smoke.

*35) Figure 2*

*3rd, not 3th. please correct the label, only 4 types should be appear here.*

The reviewer is right. The name of the layer is corrected (3rd was replaced with 3th).

*Is it pure smoke?*
*please correct the label, only 4 types should be appear here. Is it pure smoke?*

The reviewer is right. The PS label was added to the plot, that corresponds to the category:  polluted continental + smoke.

The following references have been added in the revised manuscript

---

## Referee Report (RR1)

**General comments:**

The manuscript " Comparison of two automated aerosol typing methods and their application on an EARLINET station" main goal is to compare the performance/ability of two algorithms devoted to identify the nature of aerosol layers from vertical distribution of aerosol optical intensive properties derived using LIDAR retrievals of spectral extinction and backscatter coefficients.

The Thessaloniki EARLINET station database is used to perform the test.

First, the algorithms performances are tested focusing on 3 reference cases studies of well characterized environmental scenarios, a dust transport event over Thessaloniki, a smoke dominated scenario relate to biomass burning and a multilayer scenario including different type of aerosols.

Afterward, the algorithms are tested using a larger database including 54 Lidar aerosol profiles taken from 2012 to 2015 over Thessaloniki.

The study  major conclusions are:

1) The algorithms presented similar performance for the environmental scenario regarded as PollutedSmoke, which seems to be the dominant for Thessaloniki.

2) The agreement between the two algorithms is less effective for Dust and CleanContinental aerosol scenarios, which the authors related to differences between the algorithms associated with the definitions of the typical range of intensive optical properties

Regarding the 3 reference cases studies, I would say that there is a need for a better characterization. Backward trajectories alone are not a robust to support the presence of air mass carrying a certain type of aerosol. Being the aim of these cases analysis to test the algorithms  when there is  a recognized condition of a specific type or a combination of aerosols, I would expect additional support data(satellite aod map, even fire spot map mentioned in the text) to characterize these scenarios. I'm aware that in end the aim is to test the algorithms performance from operational perspective, but that seems to be the focus of the second part of the analysis, when a larger database is used.

A major challenge of the manuscript is to emphasize results beyond those related to predictable differences resulting from recognized aspects of both methods. First, the similar performance regarding the dominant type of aerosol observed over Thessaloniki would not be expected since both methods are based in similar intensive optical properties and the range of these optical properties for this type of aerosol is closer than that from other aerosol types?

Therefore, an interesting part of the analysis would be to discuss the mismatches between the methods, which the authors also show that most of them are related to the differences between threshold range previously defined to characterize aerosol type.

From my point of view, instead of (or along) the 3 reference cases studies previously selected and discussed, it would interesting to see further analysis of cases studies selected from those mismatches cases (Figure 7), for instance, those when NATALI classified aerosol layers as PollutedSmoke while EMD named the layers as dust.

As the authors highlighted in their conclusion, indeed a test with focus on high resolution limits would be a more fruitful test to discuss the algorithms performance.

I would recommend a careful english revision targeting in particular the use of propositions.

**Summary: Two central points that I think would help to improve the manuscript:**

1) Improve the aerosol scenarios characterization for the 3 reference cases studies

2) Include further analysis regarding the mismatch between the methods, and cases studies taken from mismatches events (Figure 7) may help in this.

**Specific corrections/suggestions**

**Pag 1, Line 3:** I suggest the author to name aerosol optical properties, at least few examples, used in the study (either intensive or extensive). Throughout the abstract the author have not mention a single optical properties used in the study.

**Pag 1, Line 4:** *"... on three distinct cases..."* Need to improve the case studies introduction/description, which cases are the authors mentioning, lidar profiles or environmental scenarios? I'm aware of the cases because I've read the manuscript, but a brief idea in the abstract about the nature of the cases that the authors are talking would be important.

**Pag 5, Line 143:** *"...Color ratios - CI.."* here Color ratio is shortened to CI afterward in the text the expression CR is used. Please verify.

**Pag 6, Line 143**: *"... and changes largely for aerosols with different chemical and physical properties…"* A brief practical description of the way that chemical and physical properties of aerosol drive Lidar Ratio magnitude would be really helpful here.

**Pag 6, Line 164:** "... for 3+2 lidars.." As far as I'm aware this terms(3+2) was not clearly defined previously

**Pag 6, Line 182**: *"...between acceptable limits…"* Does this acceptable limits vary geographically or with environment scenarios? Is it possible to contextualize this limits to Thessaloniki? Why the subsequent text only describe limits for Color Ratio and Lidar Ratio?

**Pag 8, Line 249:** *"... the 5 and 4 classes…"* I'm not aware in which part of manuscript the numeric denomination of the classes was introduced.

**Pag 9, Line** 255: *"... classes that tend to reflect the same optical properties values…"* Please, clarify. Which optical properties the authors are talking?

**Pag 10, Line 294**: *"... Table 2 lists the mean aerosol optical properties…"* Suggestion: replace "*mean aerosol optical*" to " typical range of aerosol optical".

**Pag 11, Lines 328-329:** Indeed, AE and Lidar ratio between the layer 1 and 2 are similar. However, from the point of view of BAEs, the differences between both layers are not negligible. Would you comment about the possible reasons.

**Pag 13, Lines 391-390:** *"...the algorithms take into consideration different combination of the intensive optical properties…"* I would suggest the authors to think about a way(table, flow chart, diagram) to summarize the major differences between the algorithms regarding intensive optical properties combination. It would make it easy to the manuscript reader.

**Pag 13, Lines 391-395**: This discussion is really important in a way that it helps to the understanding of the limitations and improvement potential for both methods. However, is to short. I wonder if the authors could add a little more on this matter.

**Pag 14, Lines 438 - 440**: The authors seems to provide a general discussion of challenges aspects of their results (*"the reasons for the differences"*) from point of view of others studies results. But it would be interesting if they can also explore the reasons for the differences in more specific way, focusing on their data.

**Pag 14, Line 445**: *"...uncertainties of the input measurements..."* Measurements uncertainties are mentioned throughout the text as an important aspect and yet it is barely discussed.

**Pag 16, Line 505 - 525:** I think that this discussion would fit better early in the manuscript when the differences between the algorithms(EMD and NATALI) are discussed. This content is not a consequence of the results of the present study, but an intrinsic part of the algorithms that clearly drive the results obtained.

**Pag 17, Line 527:** "...of the applied thresholds…" replace to "...of the range thresholds applied to the intensive optical properties applied…"

---

## Editor Decision (ED1)

**Review of "Comparison of two automated aerosol typing methods and their application on an EARLINET station" by K. A. Voudouri, N. Siomos, K. Michailidis, N. Papagiannopoulos, L. Mona, C. Cornacchia, D. Nicolae, and D. Balis**

This paper uses EARLINET 3α + 2β lidar measurements to compare the performance of two different aerosol typing algorithms: EMD (i.e., the EARLINET Mahalanobis distance-based typing algorithm) and NATALI (the Neural network Aerosol Typing Algorithm based on LIdar data).

This is the second time I have reviewed this paper, and I'm delighted to say that the authors have done a highly commendable job of addressing all of the many remarks I made regarding their initial draft. In particular, the addition of a well-formulated set of science questions to motivate the algorithm performance comparisons now makes this paper suitable for publication in ACP.

As with my previous review, I am attaching a (very lightly) annotated version of the manuscript. I hope the authors will consider these remarks in a final draft. However, the paper could be published with only minor technical corrections, and publication should not be contingent on the authors' responses (or lack thereof) to any of my comments.

[revised manuscript text omitted]

---

## Author Response (AR2)

*This paper uses EARLINET 3α + 2β lidar measurements to compare the performance of two different aerosol typing algorithms: EMD (i.e., the EARLINET Mahalanobis distance-based typing algorithm) and NATALI (the Neural network Aerosol Typing Algorithm based on Lidar data).*
*This is the second time I have reviewed this paper, and I'm delighted to say that the authors have done a highly commendable job of addressing all of the many remarks I made regarding their initial draft. In particular, the addition of a well-formulated set of science questions to motivate the algorithm performance comparisons now makes this paper suitable for publication in ACP.*
*As with my previous review, I am attaching a (very lightly) annotated version of the manuscript. I hope the authors will consider these remarks in a final draft. However, the paper could be published with only minor technical corrections, and publication should not be contingent on the authors' responses (or lack thereof) to any of my comments.*

We thank the editor and the reviewers for their fruitful comments. Indeed, all comments have been extensively taken into account in the present form of the manuscript. In what follows, answers to comments are reported just below each related comment.

*1) how does NATALI classify the aerosol layers in the EMD training set?*

Doina et al., (2018) showed that the performance of the typing algorithm between sunthetic data and real measurements ( CALIPSO dataset), as it is shown in Figure 9 of this paper, is consistent.

*2) this is very useful information. but why do the authors say "appeared to be"? did the sensitivity analysis not provide a quantitative assessment of relative influence?*

The reviewer is right. A quantitative analysis was made to identify which classifying properties provide the adequate information to better predict the correct aerosol class. Two statistical parameters were used in the analysis: the total and the partial Wilks' lambda (3; Wilks, 1963) that are widely used, e.g, Burton et al. (2012) and Russell et al. (2014).
So, the text is modified accordingly:
"The intensive properties that hold the most weight among others in the classification are the backscatter-related Ångström Exponent at 355nm and 1064nm (BAE 355/1064), the lidar ratio at 532 nm (LR 532), and the ratio of the lidar ratios (LR 532 /LR 355)."

***3) disagree; 1.99 ± 0.13, 1.84 ± 0.06, and 1.61 ± 0.08 are really not consistent with 0.9 ± 0.26. (note: 0.9 + 2 × 0.26 = 1.42, which is less than 1.61 - 2 × 0.08. so even the closest case does not overlap within 2 standard deviations)***

The reviewer is right. Values reported by Teche et al., 2011a, are lower than the typical values of biomass burning or forest fire smoke, reported in literature (Wandinger et al., 2002; Balis et al., 2003; Muller et al., 2005; Baars et al., 2012). The sentence has been replaced with: "These values are consistent with previous reported ones, e.g., Baars et al. (2012) reported values of the extinction Ångström exponent for smoke of 1.17 ± 0.44."

***4) that are used***
The text is modified accordingly.

***3) I think you mean figure 3, both here and throughout the rest of this paragraph***
We thank the reviewer for this comment. Indeed, the whole paragraph refers to Figure 3.

***4) that's odd. according to line 374, the mean Ångström exponent is 1.72 ± 0.07, which seems very high for a marine layer (e.g., see Smirnov et al., 2002; https://doi.org/10.1175/1520-0469(2002)059<0501:OPOAAI>2.0.CO;2)***

The reviewer is right. The mean Ångström exponent is rather high for marine particles, but this layer is typed by NATALI as Maritime one, due to the low Lidar Ratio values (29±0.77 sr for 355nm and 30±0.91 sr for 532nm), as NATALI uses the full information of the intensive optical properties of each layer.

***5) unclear; do you mean "that prohibited ANNs from returning an aerosol type classification"?***
The reviewer is right. The sentence has been replaced with the following one: "..that prohibited ANNs from returning an aerosol type classification."

***6) too sensitive, perhaps? I'm thinking here of the marine case in figure 3.***
Indeed, in general, NATALI software seems more sensitive to the Maritime category. See comment 4.

***7)though***
The sentence has been modified to:"...even though both classifications schemes have been trained using supervised learning techniques."

***8) rephrase? requiring "perfect" calibration might be asking a bit too much***
The reviewer is right. The perfect calibration can be rather ambitious. Therefore, we modified the sentence, which now reads: "Both algorithms are unable to return

aerosol type for not well calibrated input data, i.e., when one or more intensive optical properties have large uncertainties."

**9) consistent**
The text has been modified to: "This result is consistent with the CALIPSO scheme at global level and with results about Thessaloniki site characterization done by Siomos et. al (2018b)."

**10) how well successful is NATALI when it attempts to classify the EMD training set?**
See comment 1.

**11) unclear; do you mean "to confirm the inferred aerosol type"?**

[revised manuscript text omitted]